# Cell differentiation defines acute and chronic infection cell types in *Staphylococcus aureus*

Juan-Carlos García-Betancur[1,2], Angel Goñi-Moreno[3], Thomas Horger[4], Melanie Schott[5], Malvika Sharan[1], Julian Eikmeier[1,2], Barbara Wohlmuth[4], Alma Zernecke[5], Knut Ohlsen[1], Christina Kuttler[4], Daniel Lopez[1,2,6]*

[1]Institute for Molecular Infection Biology, University of Würzburg, Würzburg, Germany; [2]Research Center for Infectious Diseases, University of Würzburg, Würzburg, Germany; [3]School of Computing Science, Newcastle University, Newcastle, United Kingdom; [4]Department of Mathematics, Technical University of Munich, Garching, Germany; [5]Institute of Clinical Biochemistry and Pathobiochemistry, University Hospital Würzburg, Würzburg, Germany; [6]National Center for Biotechnology, Madrid, Spain

**Abstract** A central question to biology is how pathogenic bacteria initiate acute or chronic infections. Here we describe a genetic program for cell-fate decision in the opportunistic human pathogen *Staphylococcus aureus*, which generates the phenotypic bifurcation of the cells into two genetically identical but different cell types during the course of an infection. Whereas one cell type promotes the formation of biofilms that contribute to chronic infections, the second type is planktonic and produces the toxins that contribute to acute bacteremia. We identified a bimodal switch in the *agr* quorum sensing system that antagonistically regulates the differentiation of these two physiologically distinct cell types. We found that extracellular signals affect the behavior of the *agr* bimodal switch and modify the size of the specialized subpopulations in specific colonization niches. For instance, magnesium-enriched colonization niches causes magnesium binding to *S. aureus* teichoic acids and increases bacterial cell wall rigidity. This signal triggers a genetic program that ultimately downregulates the *agr* bimodal switch. Colonization niches with different magnesium concentrations influence the bimodal system activity, which defines a distinct ratio between these subpopulations; this in turn leads to distinct infection outcomes in vitro and in an in vivo murine infection model. Cell differentiation generates physiological heterogeneity in clonal bacterial infections and helps to determine the distinct infection types.

DOI: https://doi.org/10.7554/eLife.28023.001

*For correspondence:
dlopez@cnb.csic.es

Competing interests: The authors declare that no competing interests exist.

## Introduction

Nosocomial pathogens often cause a broad range of diseases using diverse virulence factors, such as production of tissue-damaging toxins or production of adhesins during biofilm formation (*Bush et al., 2011*). *Staphylococcus aureus* is one such pathogen that is able to cause different types of life-threatening infections in hospital settings, from acute bacteremia to endocarditis, pneumonia and chronic biofilm-associated infections in prosthetic devices (*Otto, 2012*). The underlying cellular processes that enable *S. aureus* to provoke these disparate types of infections is likely driven by host-microbe interactions (*Casadevall et al., 2011*), in which specific, yet-to-be-described extracellular signals play a role to generate distinct, locally defined types of infections (*Veening et al., 2008*; *López and Kolter, 2010*). Determining the cellular processes and the nature of the extracellular

**eLife digest** While in hospital, patients can be unwittingly exposed to bacteria that can cause disease. These hospital-associated bacteria can lead to potentially life-threatening infections that may also complicate the treatment of the patients' existing medical conditions. *Staphylococcus aureus* is one such bacterium, and it can cause several types of infection including pneumonia, blood infections and long-term infections of prosthetic devices.

It is thought that *S. aureus* is able to cause so many different types of infection because it is capable of colonizing distinct tissues and organs in various parts of the body. Understanding the biological processes that drive the different infections is crucial to improving how these infections are treated.

*S. aureus* lives either as an independent, free-swimming cell or as part of a community known as a biofilm. These different lifestyles dictate the type of infection the bacterium can cause, with free-swimming cells producing toxins that contribute to intense, usually short-lived, infections and biofilms promoting longer-term infections that are difficult to eradicate. However, it is not clear how a population of *S. aureus* cells chooses to adopt a particular lifestyle and whether there are any environmental signals that influence this decision.

Here, Garcia-Betancur et al. found that *S. aureus* populations contain small groups of cells that have already specialized into a particular lifestyle. These groups of cells collectively influence the choice made by other cells in the population. While both lifestyles will be represented in the population, environmental factors influence the numbers of cells that initially adopt each type of lifestyle, which ultimately affects the choice made by the rest of the population. For example, if the bacteria colonize a tissue or organ that contains high levels of magnesium ions, the population is more likely to form biofilms.

In the future, the findings of Garcia-Betancur et al. may help us to predict how an infection may develop in a particular patient, which may help to diagnose the infection more quickly and allow it to be treated more effectively.

DOI: https://doi.org/10.7554/eLife.28023.002

signals that define the different infection outcomes is crucial for understanding how difficult-to-treat bacterial infections develop and for improving strategies to overcome antimicrobial resistance.

In *S. aureus*, infection outcome is controlled by the *agr* quorum sensing program, which is autoactivated in response to the self-produced extracellular signal AIP (<u>a</u>uto<u>i</u>nducing <u>p</u>eptide) (*Recsei et al., 1986*). AIP binds to the AgrC histidine kinase membrane receptor and activates its cognate regulator AgrA via phosphorylation (*Figure 1A*). AgrA~P induces changes in cellular gene expression that results in rapid bacterial dispersion in the host and acute bacteremia (*Thoendel et al., 2011*). Dispersion of *S. aureus* requires upregulation of surfactant phenol-soluble modulins (*psmα* and *psmβ*), which are amphipathic small peptides that contribute to bacteria detachment (*Li et al., 2009a*; *Peschel and Otto, 2013*) and destabilize cell membranes, rendering them cytotoxic to host cells. Modulins are usually expressed during acute infections, as well as hemolytic toxins (*hla*, *hlb*, *hlg*) that facilitate tissue disruption during septicemia (*Recsei et al., 1986*). In contrast, *agr* activation indirectly downregulates the *icaADBC* operon genes needed to synthesize the extracellular polysaccharide matrix that protects cells within a biofilm (PNAG or PIA), as well as several adhesion proteins (SpA and other MSCRAMM proteins) responsible for cell aggregation/attachment during biofilm formation (*Recsei et al., 1986*; *Boles and Horswill, 2008*; *Peng et al., 1988*). Biofilms, which are associated with untreatable chronic infections, protect bacteria from antibiotics and host defenses (*Lewis, 2008*; *Lopez et al., 2010*; *Nadell et al., 2009*; *Parsek and Singh, 2003*). The *S. aureus agr* quorum sensing system antagonistically regulates the activation of planktonic and biofilm-associated lifestyles (*Recsei et al., 1986*; *Boles and Horswill, 2008*; *Peng et al., 1988*), which contribute to the development of acute and chronic infection outcomes, respectively. A large number of positive and negative regulators controls *agr* expression. Among those, the *agr* system is inhibited by the σ$^B$ sigma factor (*Bischoff et al., 2001*). Activation of σ$^B$ occurs during early stationary phase (*Senn et al., 2005*) in response to distinct types of cellular stresses (*Geiger et al., 2014*; *Geiger and Wolz, 2014*; *Kästle et al., 2015*). σ$^B$ triggers a general

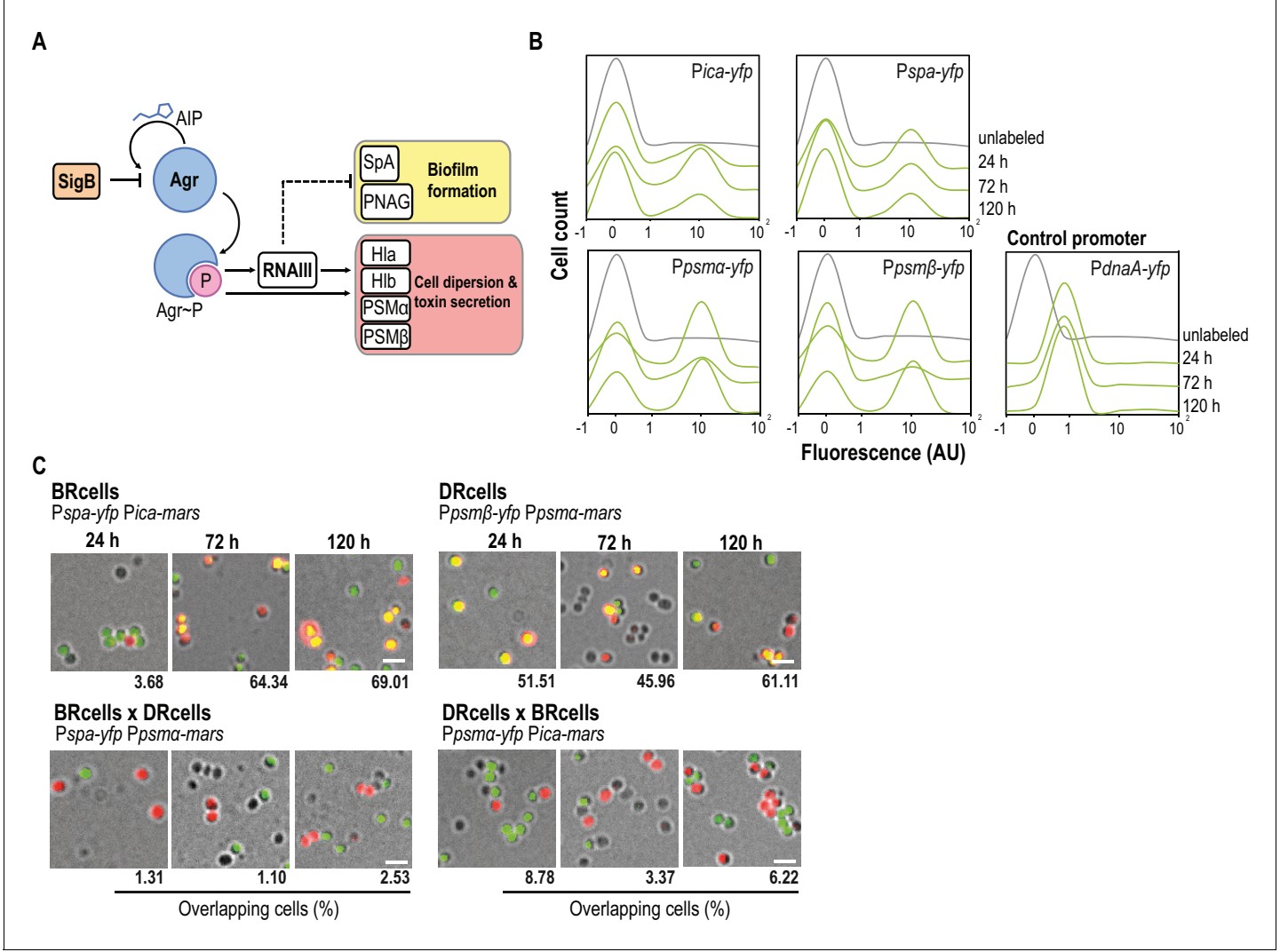

**Figure 1.** *Staphylococcus aureus* aggregates contain specialized cell types. (**A**) Scheme of the *agr* signaling cascade in *S. aureus*. AIP binds to the AgrC histidine kinase membrane receptor and activates its cognate regulator AgrA via phosphorylation (AgrA~P). AgrA~P upregulates toxin-coding genes that are also responsible for cell dispersion (*hla, hlb, psmα* and *psmβ*) and it downregulates genes involved in biofilm formation (*ica* and *spa*). (**B**) Quantitative analysis of fluorescence microscopy images of *agr*-related promoters. The control promoter is the *agr*-independent *dnaA*, which has a monomodal expression pattern. We counted 700 random cells from each of three independent microscopic fields from independent experiments (2100 cells total for each strain). (**C**) Fluorescence microscopy images of double-labeled cells at various times during aggregate formation (24, 72 and 120 hr). Percentages of cells quantified in the fields with positive overlapping signal. Cells were counted as in (**A**). A signal is considered to overlap when signals are detected in a 3:1-1:3 range, the range in which green and red signals merge to yellow. Top row, double-labeled strains with P*ica-yfp*, P*spa-yfp* (BRcells) and P*psmα-yfp*, P*psmβ-yfp* (DRcells) promoters. Bottom row, double-labeled strains with antagonistic promoters. Bar = 2 μm.

DOI: https://doi.org/10.7554/eLife.28023.003

The following figure supplement is available for figure 1:

**Figure supplement 1.** *Staphylococcus aureus* multicellular aggregates contain distinct cell types.

DOI: https://doi.org/10.7554/eLife.28023.004

stress response that affects expression of a number of virulence and stress-response genes and indirectly represses *agr* (*Thoendel et al., 2011*). Thus, σ[B] antagonizes the influence of the *agr* system on virulence factor expression (*Senn et al., 2005*; *Pané-Farré et al., 2006*) and biofilm formation (*Bischoff et al., 2001*; *Kullik et al., 1998*).

This understanding of *agr*-mediated antagonistic regulation of chronic and acute *S. aureus* infection outcomes was built on comparative analyses of clinical isolates and characterization of infection

related mutants. Through this approach, *agr*-defective isolates are frequently identified from chronic infections as these mutants usually show reduced hemolytic activity and develop robust biofilms (*Fischer et al., 2014*; *Goerke and Wolz, 2010*; *Grundmeier et al., 2010*; *Hirschhausen et al., 2013*; *Savage et al., 2013*). In addition, *agr* dysfunction is frequently correlated with chronic persistent *S. aureus* infections (*Fowler et al., 2004*) such as small colony variants (SCV). SCV have exceptionally low *agr* expression levels (*Kahl et al., 2016*) and high expression of biofilm-related genes (*Tuchscherr et al., 2010*). High $\sigma^B$ expression is important for SCV phenotype acquisition (*Mitchell et al., 2013*), because *sigB* mutants do not generate SCV (*Tuchscherr and Löffler, 2016*; *Tuchscherr et al., 2015*). However, whether the capacity of nosocomial pathogens, such as *S. aureus,* to cause distinct types of infections is restricted to the emergence of genetic variants is still unclear. *Staphylococcus aureus* cells are exposed to a variety of local environmental signals during the course of an infection that can influence bacterial gene expression and thus, their infective potential in a given infection niche. These signals include, but are not limited to, changes in nutrient availability, temperature, pH, osmolarity or, oxygen concentration. *Staphylococcus aureus* might be able to respond collectively to these extracellular cues to adapt its behavior in a fluctuating environment (*Münzenmayer et al., 2016*), allowing staphylococcal communities to generate distinct, locally defined infection types without modification of the bacterial genome (*Veening et al., 2008*; *López and Kolter, 2010*).

It has been hypothesized that changes in bacterial virulence potential are a response to local concentrations of tissue-specific signals, which have an important role in determining infection outcome (*Cheung et al., 2004*). Yet how bacteria prepare for such unpredictable environmental changes is a question that remains unanswered. A fundamental feature of microbial cells is their ability to adapt to diverse environmental conditions by differentiating into specialized cell types (*Arnoldini et al., 2014*; *López et al., 2009*; *Veening et al., 2005*). In most cases, the extracellular cues are responsible for defining coexisting cell fates in bacterial populations. Cell fates are genetically identical and phenotypically distinct bacterial subpopulations that express heterogeneously different sets of genes and have distinct functions within the microbial community (*George et al., 2015*). A classical example of this is the bacterial response to antibiotics. Antibiotics kill most *S. aureus* cells, but it is frequent to observe a small subpopulation of genetically identical but antibiotic-persister cells that can cause recurrent infections in a post-antibiotic period (*Bigger, 1944*; *Lewis and cells, 2007*).

The relative simplicity of *agr*-mediated antagonistic regulation of planktonic and biofilm-associated lifestyles provides a natural model to analyze how *S. aureus* cells collectively establish acute or chronic infection lifestyles and to identify extracellular factors that influence activation of the cellular program that leads to prevalence of one infection program over the other. Here, we report a bimodal behavior in the *agr* quorum sensing system that antagonistically regulates the differentiation of two genetically identical but physiologically distinct specialized cell types in *S. aureus* communities. One cell type contributes to the formation of biofilms responsible for chronic infections, whereas a second was constituted by dispersed cells that produced the toxins that contribute to an acute bacteremia. These subpopulations were present in *S. aureus* communities at different ratios depending on growth conditions, which contributed to determining the outcome of infection. We found that colonization niches with higher $Mg^{2+}$ concentrations, which is inherent in tissues colonized preferentially by *S. aureus* (*Günther, 2011*; *Jahnen-Dechent and Ketteler, 2012*), influenced the bimodal switch and increased the size of the subpopulation of cells specialized in biofilm formation, as $Mg^{2+}$ binding to teichoic acids increases cell wall rigidity and triggered a $\sigma^B$ stress-induced genetic cascade that downregulates *agr*. In a mouse model, bacterial cell differentiation occurred during in vivo infections and the $Mg^{2+}$ concentration in infected organs influenced collective bacterial behavior in simultaneous progress to a biofilm-associated chronic infection or a disperse bacteremia. This study shows that cell differentiation in *S. aureus* helps to diversify the types of infections that arise simultaneously from an infection caused by a clonal population of bacteria.

## Results

### *Staphylococcus aureus* multicellular aggregates differentiate cell types

We explored the role of *agr*-mediated antagonistic regulation of planktonic and biofilm-associated lifestyles in *S. aureus* aggregates growing on $Mg^{2+}$-enriched TSB medium (TSBMg), in which most *S.*

*aureus* clinical isolates develop robust multicellular aggregates (*Koch et al., 2014*). To study biofilm gene expression, we introduced transcriptional fusions of biofilm-associated *ica/spa* genes. The *ica* operon is responsible for production of the exopolysaccharide polymeric matrix (PNAG or PIA) that lends consistency to the biofilm. The *spa* gene encodes a cell wall-anchored adhesion protein, adhesin that is responsible for *S. aureus* cell aggregation and attachment to surfaces during biofilm formation (*Recsei et al., 1986*; *Boles and Horswill, 2008*; *Peng et al., 1988*). To monitor planktonic gene expression, we generated transcriptional fusions of *psmα* and *psmβ* genes. These genes code for small peptides, the phenol-soluble modulins, whose expression depends directly on *agr*. Due to their surfactant properties, PSMα and PSMβ act as cytolytic toxins that contribute to bacterial dispersion and play an important role in acute staphylococcal infections (*Li et al., 2009a*; *Peschel and Otto, 2013*) (*Figure 1A*). These reporters were introduced into neutral loci in the *S. aureus* chromosome to ensure expression of a single reporter copy in each cell (*Yepes et al., 2014*); we monitored their expression at the single-cell level in *S. aureus* aggregates using quantitative analysis of fluorescence microscopy images (*Figure 1B* and *Figure 1—figure supplement 1A–B*). All reporters showed bimodal expression and indicated the bifurcation of two cell subpopulations in *S. aureus* aggregates, one with lower and another with higher fluorescence levels. This bimodal expression pattern differed from the unimodal expression of the housekeeping and *agr*-independent gene *dnaA*, used as control reporter (*Figure 1B* and *Figure 1—figure supplement 1A*). Cultures of different *S. aureus* isolates showed bimodal expression of these reporters (*Figure 1B* and *Figure 1—figure supplement 1C*), which suggests that cell differentiation is a general phenomenon in *S. aureus*.

Monitoring the temporal dynamics of the subpopulations that bifurcated during the development of the microbial aggregates from an initial inoculum in TSBMg revealed larger subpopulations of *ica*- and *spa*-expressing cells in early developmental stages (~72 hr; *Figure 1B*), compared to the size of these subpopulation at later growth stages (~120 hr). In contrast, the size of the subpopulations of *psmα/β*-expressing cells increased concomitantly with time, consistent with the reported antagonistic regulation of *ica/spa* and *psmα/β* by *agr* (*Recsei et al., 1986*; *Boles and Horswill, 2008*; *Peng et al., 1988*). We generated strains labeled with different pairwise combinations of these reporters, which were both introduced into neutral loci in the *S. aureus* chromosome; this allowed quantitative analysis of fluorescence microscopy images to examine simultaneous expression in *S. aureus* microbial communities (*Figure 1C* and *Figure 1—figure supplement 1D*). This approach indicated coexpression of *ica* with *spa* and of *psmα* with *psmβ* in two distinct cell subpopulations, showing the bifurcation of two distinct subpopulation of cells specialized in expressing *ica, spa* and other biofilm-related genes (BRcells) and cells expressing *psmα/β* dispersion-related genes (DRcells).

## Differential AgrA~P affinity to P2 and P3 promoters generates the *agr* positive feedback loop that differentiates BRcells and DRcells

The differential expression of *agr*-related genes in the distinct cell types led us to analyze the molecular mechanism of *agr*-mediated cell differentiation. The *agr* system is autoactivated once the extracellular AIP concentration reaches a given threshold (10–14 µM) (*MDowell et al., 2001*) and is inhibited by σ[B] induction (*Bischoff et al., 2001*). Following *agr* activation, AgrA~P directly upregulates *psmα/β* expression (*Queck et al., 2008*) and binds to the two adjacent divergent promoters P2 and P3, which trigger expression of RNAII and RNAIII transcripts, respectively (*Koenig et al., 2004*). RNAII upregulates the *agrBDCA* operon, which encodes the *agr* signal transduction cascade, including the AIP signal, the AgrC sensor kinase and its AgrA cognate regulator. Therefore, AgrA~P binding to the P2 promoter constitutes a positive feedback loop in which AgrA~P regulator induces expression of the *agrBDCA* operon, which encodes the entire *agr* signal transduction cascade (*Thoendel et al., 2011*; *Queck et al., 2008*). Bimodal gene expression in microbial populations is usually triggered by a positive feedback loop in which a gene product induces its own expression. We hypothesize that, once a certain AgrA~P threshold is reached in a cell, AgrA~P induces its own expression and these cells maintain high AgrA~P levels. Thus, AgrA~P and AgrA-controlled genes will thus be activated in that cell, including upregulation of RNAIII via activation of the P3 promoter. RNAIII positively regulates a pool of *agr*-dependent genes that encode the cytotoxic toxins and virulence factors responsible for acute infection (*Koenig et al., 2004*). Activation of the P3 promoter leads DRcells to specialize in dispersion and virulence. In contrast, cells that do not achieve the AgrA~P expression threshold needed to induce the positive feedback mechanism will not induce P3 promoter expression. In these cells, genes normally repressed by AgrA~P will be upregulated,

including biofilm-related genes, which licenses cells to differentiate as biofilm-producing BRcell types.

To determine whether activation of the *agr* positive feedback loop is sufficient to generate bimodality in a bacterial population, we genetically engineered an orthogonal *agr* system in *B. subtilis* in which the *agr* positive feedback loop was isolated from its native complex regulatory network (*Audretsch et al., 2013*) and thus exempt from interference from additional staphylococcal regulatory inputs. In this orthogonal system, *B. subtilis* harbored $P_{psm\alpha}$-*yfp* or $P_{psm\beta}$-*yfp* reporters and expressed the membrane kinase AgrC and its cognate regulator AgrA under the control of the AgrA-inducible P2 promoter (*Figure 2A*). The orthogonal system does not express staphylococcal $\sigma^B$, and $\sigma^B$ from *B. subtilis* did not interfere with the *agr* system, since we detected similar reporter expression in wild type (WT) and $\Delta sigB$ strains (*Figure 2—figure supplement 1A*), ensuring the absence of the *agr* inhibitory input signal. We used this orthogonal system to identify the minimal components necessary for bimodal expression of *agr*-related genes, using *psmα/β* expression as readout for *agr* activity (*Figure 2B*) (*Zhang et al., 2015*). Activation of the *agr* positive feedback loop in the orthogonal system requires addition of purified AIP (10 µM) to *B. subtilis* cultures (AIP contains a thiolactone ring thus it cannot be synthetically produced). AIP addition activates AgrC sensor kinase, which phosphorylates the AgrA regulator. The AgrA~P active form binds the P2 promoter to express high AgrA~P levels and activate the positive feedback loop; this resulted in bimodal expression of chromosomally integrated $P_{psm\alpha}$-*yfp* or $P_{psm\beta}$-*yfp* reporters, with a cell subpopulation in which reporter expression increased during a transition in which cells switched from off to on. In contrast, chromosomally integrated $P_{ica}$-*yfp* and $P_{spa}$-*yfp* reporters, which are not controlled directly by the AgrA~P, did not activate in response to added AIP. In a similar manner, we did not detect fluorescence in control experiments with strains lacking the AgrC-AgrA system, or when no AIP was added (*Figure 2B* and *Figure 2—figure supplement 1A*), demonstrating that stochastic expression of these reporter genes does not account for bifurcation of the cell populations. These results indicate that the minimal *agr* genetic program harbored in the orthogonal system acts as an autonomous program for cell differentiation in bacterial populations.

Identification of the molecular mechanism that leads *agr* to act as an autonomous program to define cell fate bifurcation in *S. aureus* required analysis of the *agr*-signaling cascade in the presence of AIP. We therefore generated two additional orthogonal systems to monitor P2 ($P_{RNAII}$-*yfp*) and P3 ($P_{RNAIII}$-*yfp*) activation independently (*Figure 2C*). In response to exogenous AIP (10 µM), these systems showed a transition period during which P2 and P3 cells switched on after which subpopulation sizes remained constant. The P2-expressing cell subpopulation differentiated earlier, however, and showed a more intense fluorescence signal in a larger cell subpopulation over time compared to the orthogonal system that differentiated P3-expressing cells. This result suggests that P2 promoter is more sensitive than P3 to *agr* activation, a characteristic feature of positive feedback loops in bimodal systems (*Siebring et al., 2012*). Based on these results, we hypothesized that the bimodal behavior of *agr* and thus, cell differentiation in *S. aureus*, relies on the differential affinity of AgrA~P for P2 and P3 promoters. P2 thus activates the feedback loop at lower AgrA~P concentrations and only in a subpopulation of cells. These cells contain high AgrA~P levels, which licenses them to trigger the less-sensitive P3 promoter and induce the *agr* regulon, leading the cells to specialize in dispersion and virulence and become DRcells. Cells that express P2 below the threshold cannot activate the *agr* feedback loop and are thus unable to induce P3 promoter expression. In this subpopulation, *agr*-repressed genes are upregulated, including genes involved in biofilm formation, which licenses them to differentiate as BRcells.

We tested this hypothesis by first analyzing the dynamics of the *agr* positive feedback loop using mathematical modeling coupled to computational simulations (*Figure 2—figure supplement 1B–E*) (*Chong et al., 2014*; *Golding et al., 2005*). The full network dynamics remained constant, since the orthogonal system was similar for all promoters tested. We optimized the promoter-specific rates; $K_{on}$ (association) and $K_{off}$ (dissociation) denote interaction with AgrA~P and $K_t$ denotes reporter transcription (*Figure 2—figure supplement 1B*) (*Goñi-Moreno et al., 2016*). The correct combination of these promoter rates sufficed to explain bimodal fluorescence distribution. Although the simulations considered AgrA~P saturation, we consistently detected bimodal reporter expression, with on and off cell subpopulations always present. Simulations favored a model in which P2-activated positive feedback loop induces P3-driven bimodal expression in response to AIP concentration or

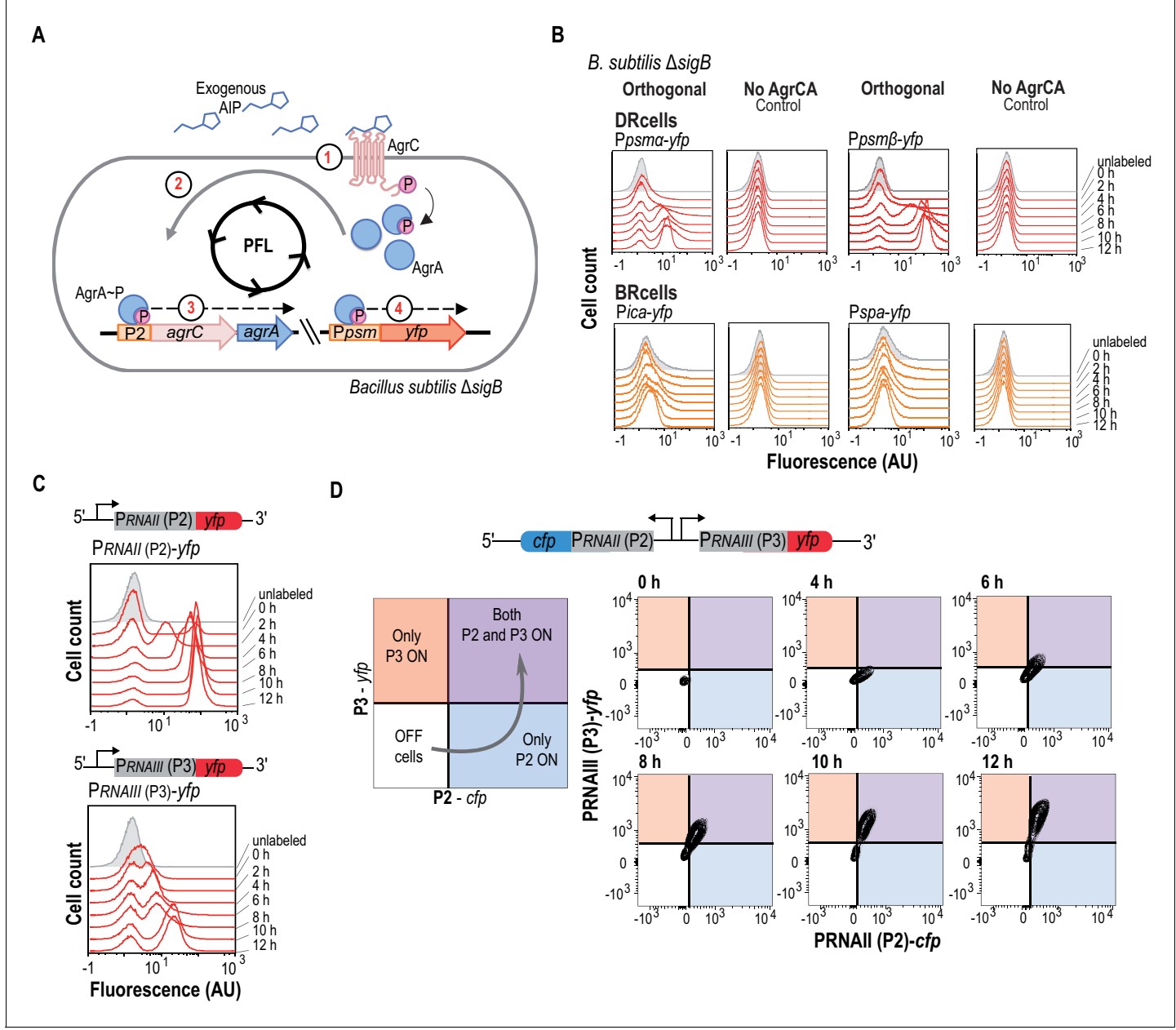

**Figure 2.** The *agr QS* system is an autonomous program for cell fate decision-making. (**A**) Scheme showing the orthogonal system generated in *Bacillus subtilis*. In this system, the membrane kinase AgrC and its cognate regulator AgrA were expressed under the control of their own P2 promoter. This strain also harbors a P*psmα-yfp* or a P*psmβ-yfp* reporter, which allows monitoring *agr* activation. Addition of purified AIP to *B. subtilis* cultures as input signal activated the *agr* system and led to bimodal expression of chromosomally integrated P*psmα-* or P*psmβ-yfp* reporters. (**B**) Flow cytometry profiles of the orthogonal *agr* system in *B. subtilis* showing bimodal expression of the *agr*-dependent *psmα/β* reporters at different times after AIP induction. Control panels show no expression of these reporters without the AgrCA chimera. The *ica/spa* promoters were not activated, as they are not regulated directly by AgrA~P (***Recsei et al., 1986***; ***Boles and Horswill, 2008***; ***Peng et al., 1988***). Cultures were grown in liquid LB medium and incubated (37°C, 12 hr, 200 rpm agitation). (**C**) Flow cytometry showed bimodal expression of P2 (up) and P3 (down) promoters after AIP induction. Cultures were grown in liquid LB medium and incubated (37°C, 12 hr, 200 rpm agitation). (**D**) Flow cytometry monitoring simultaneous expression of P$_{RNAII}$ or P2 (y-axis) and P$_{RNAIII}$ or P3 (x-axis) in a population of P2-*cfp* P3-*yfp* double-labeled cells cultured with AIP (10 μM). Samples were collected at various times and represented in a 2D graph (x axis, CFP signal; y axis, YFP signal). Dual system at various times after AIP induction. Isolines in the graph represent cell populations. The subpopulation that initially expressed the P2-*cfp* reporter was the same as that which later expressed the P3-*yfp* reporter.
DOI: https://doi.org/10.7554/eLife.28023.005

The following figure supplement is available for figure 2:

**Figure supplement 1.** Mathematical simulations of the *agr* orthologous system.

*Figure 2 continued on next page*

Figure 2 continued

DOI: https://doi.org/10.7554/eLife.28023.006

autoactivation time (*Figure 2—figure supplement 1C–E*), suggesting that DRcells resulted from sequential P2 and P3 promoter activation.

We tested this model experimentally in a dual orthogonal system harboring P2 (P$_{RNAII}$-*cfp*) and P3 (P$_{RNAIII}$-*yfp*) reporters expressed as two adjacent transcriptional units transcribed in opposite directions, similar to the chromosomal organization in the *S. aureus* genome (*Figure 2D* and *Figure 2—figure supplement 1F*). We used flow cytometry analysis with simultaneous detection of CFP and YFP signals to determine quantitatively whether the P2-expressing subpopulation becomes P3-expressing cells over time after AIP addition (10 μM). At 4 hr post-AIP induction, we detected a cell subpopulation that expressed P2; a fraction of this subpopulation activated P3 at later times (6 hr). The subpopulation of P3-expressing cells increased over time until it expressed P2 and P3 promoters uniformly. Cells that expressed only the P3 promoter were not detected. This is consistent with our hypothesis that P2-mediated activation of the *agr* positive feedback loop is necessary to increase AgrA~P levels, which in turn induces expression of the less-sensitive P3 promoter in these cells. The molecular mechanism for bimodal gene expression thus relies on the differential AgrA~P affinity for P2 and P3 promoters. P2 is very sensitive and triggers the *agr* positive feedback loop, whereas P3 induces expression of virulence genes and is necessary for DRcell specialization. In the following section, we used this information to demonstrate that the self-regulatory activity of AgrA~P via binding to the P2 promoter is essential for triggering *S. aureus* cell differentiation while other additional cues that feed into the *agr* switch only modulate the activity of the system.

## Increase in cell wall rigidity activates σ$^B$, repressing the *agr* positive feedback loop

Once the *agr* switch responsible for BRcell and DRcell differentiation is activated, distinct extracellular cues can arise from the niche to feed into the *agr* bimodal switch and modulate its activity. For instance, BRcell and DRcell subpopulations are detected at different ratios in TSB and TSBMg cultures. We hypothesized that variations in extracellular input signals would affect *agr* bimodal behavior and produce distinct outcomes in the bimodal system. This would define a distinct DRcell:BRcell ratio, which could have important clinical implications for the definition of infection outcomes. As the difference between TSB and TSBMg media resides in Mg$^{2+}$ supplementation, we tested the effect of extracellular Mg$^{2+}$ on the response of the *agr*-repressor σ$^B$, which is activated by environmental stresses. In TSBMg medium, qRT-PCR analysis showed increased expression of the σ$^B$-dependent stress gene *asp23* (<u>a</u>lkaline <u>s</u>hock <u>p</u>rotein) (*Gertz et al., 2000*) (*Figure 3A*) and of staphyloxanthin, the pigment that gives *S. aureus* its typical yellow color and whose expression is regulated directly by σ$^B$ (*Gertz et al., 2000*; *Giachino et al., 2001*) (*Figure 3—figure supplement 1A*). TSBMg medium also induced biofilm formation (*Figure 3B*). Biofilm formation likely occurred via *agr* inhibition because the Δ*sigB* strain did not form biofilm and the biofilm formation phenotype was partially recovered in a Δ*sigB*Δ*agr* double mutant (*Figure 3C*). Thus, the Mg$^{2+}$ signaling cascade acts on *agr* downregulation via σ$^B$ activation to increase BRcell subpopulation size. This is consistent with the fact that biofilm-associated *S. aureus* colonization generally occurs in Mg$^{2+}$-enriched niches such as bone and kidney, in which chronic staphylococcal infections often develop (*Günther, 2011*; *Jahnen-Dechent and Ketteler, 2012*; *Elin, 2010*). By contrast, tissues unintentionally depleted of Mg$^{2+}$ are prone to acute staphylococcal infections, as Mg$^{2+}$ sequestration from tissues due to tampon use was associated with an outbreak of staphylococcal toxic shock syndrome in women in the USA (*Parsonnet et al., 1996*; *Schlievert, 1985*).

We were prompted to analyze the molecular mechanism whereby extracellular Mg$^{2+}$ regulates the *agr* bimodal switch and increases the BRcell subpopulation. Biofilms occur in TSB supplemented with Mg$^{2+}$ but not with other cations (*Koch et al., 2014*), which suggested that Mg$^{2+}$ is a specific extracellular trigger for BRcell differentiation. Mg$^{2+}$ has a function in stabilizing the Gram-positive bacterial cell wall, which is decorated with phosphate-rich teichoic acids (TA) that contribute to membrane integrity (*Heptinstall et al., 1970*). To alleviate electrostatic repulsive interactions between neighboring phosphates, TA preferentially bind Mg$^{2+}$ cations, to form a consolidated

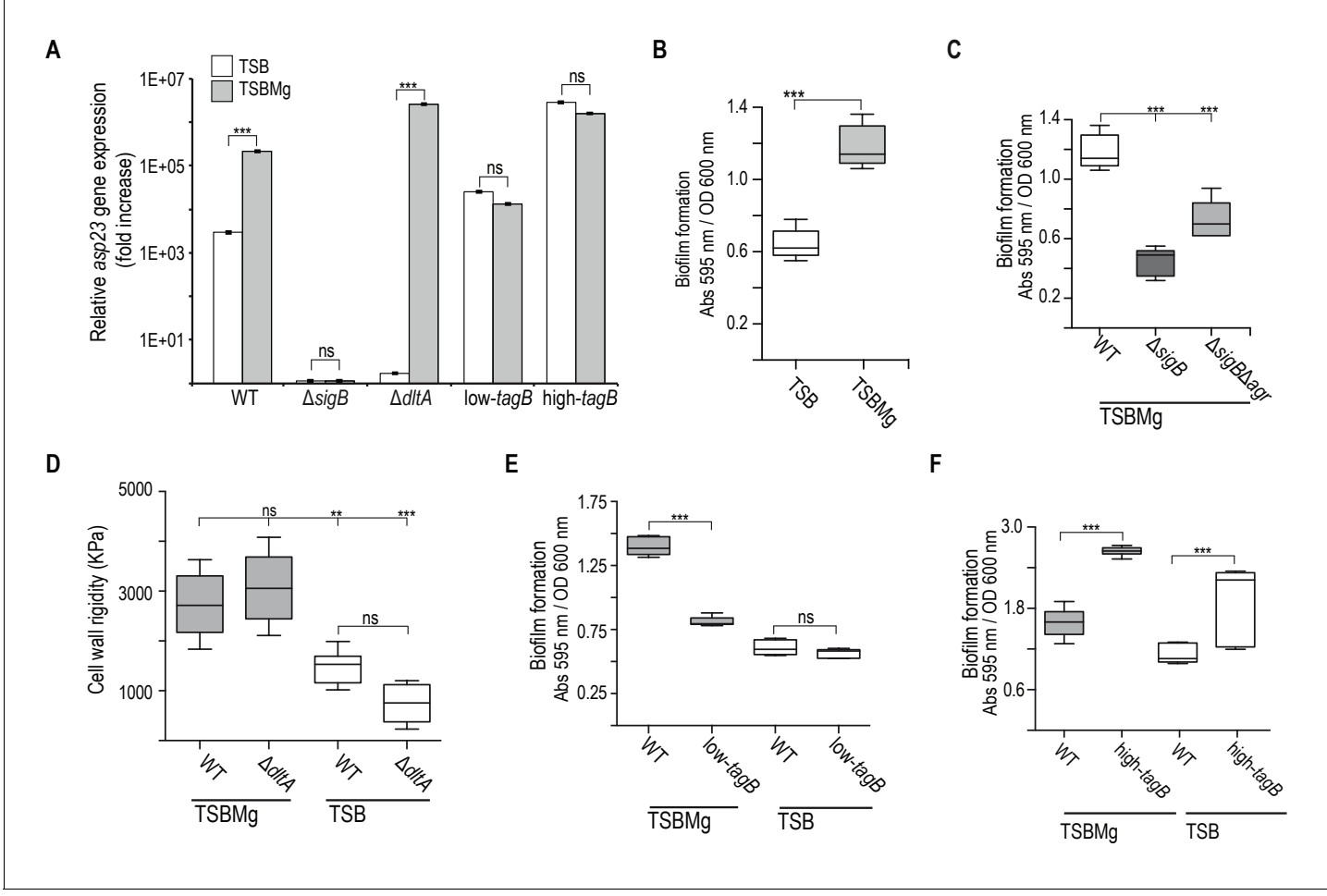

**Figure 3.** Extracellular Mg$^{2+}$ increases cell wall rigidity in *S. aureus*. (A) qRT-PCR assay to monitor σ$^B$ activation using relative *asp23* expression as readout (expression of this gene is dependent on σ$^B$ activity) in TSB and TSBMg cultures of *S. aureus* strains of different genetic backgrounds. The Δ*dltA* mutant was used as control strain to show reduced *asp23* expression in TSB and increased expression in TSBMg. (B) Biofilm formation quantification in *S. aureus* strain Newman wild type strain using the traditional microtiter assay (*O'Toole and Kolter, 1998b*) in liquid TSBMg and TSB. (C) Biofilm formation quantification in different *S. aureus* genetic backgrounds using the traditional microtiter assay (*O'Toole and Kolter, 1998b*) in liquid TSBMg and TSB. The Δ*sigB* strain did not form biofilm in TSBMg and the biofilm formation phenotype was partially recovered in a Δ*sigB*Δ*agr* double mutant. (D) Atomic force microscopy quantification of *S. aureus* cell surface rigidity (in KPa). Mean surface rigidity was measured using force-indentation curves and Young's modulus. Best fits were produced with a modified Hertz model, assuming conical punch probe geometry. The Δ*dltA* mutant serves as positive control, as described (*Saar-Dover et al., 2012*). In this mutant, D-alanine esterification of TA is absent (*Perego et al., 1995*). D-alanylation of TA introduces positively charged amines and prevents repulsive interactions between neighboring ribitol phosphates, which increases cell wall rigidity, similar to the effect of Mg$^{2+}$ incorporation to the cell wall. Cell wall rigidity was therefore compromised in the Δ*dlt* mutant when grown in TSB medium. (E, F) Quantification of biofilm formation in liquid TSBMg and TSB of *S. aureus* WT, low-*tagB* (E) and high-*tagB* strains (F). All experiments show the mean ±SD for three independent experiments (*n* = 3). Statistical significance was measured using unpaired Student's t-test for panel (A); for remaining panels, we used one-way ANOVA with Tukey's test for multiple comparisons. *p<0.05, **p<0.01, ***p<0.001; ns, no significant differences.

DOI: https://doi.org/10.7554/eLife.28023.007

The following figure supplement is available for figure 3:

**Figure supplement 1.** Extracellular Mg$^{2+}$ activates σ$^B$ stress regulon in *S. auresus*.
DOI: https://doi.org/10.7554/eLife.28023.008

network that strengthens cell envelope rigidity (*Heckels et al., 1977*; *Lambert et al., 1975a*; *Swoboda et al., 2010*). We therefore hypothesized that Mg$^{2+}$ in TSBMg stabilizes *S. aureus* TA and increases cell wall rigidity, which cues σ$^B$ activation. We tested this hypothesis using atomic force microscopy (AFM) to monitor *S. aureus* cell wall structural rigidity in vivo, comparing single cells grown in TSB and TSBMg media (*Figure 3D*) (*Saar-Dover et al., 2012*; *Touhami et al., 2004*). AFM

detects forces acting between a sharp nanoscale cantilever and the bacterial cell wall; after pressure, the cantilever deflects and force can be quantified (*Dufrêne, 2014*; *Formosa-Dague et al., 2016*). We detected greater rigidity in cells grown in TSBMg medium than those grown in TSB medium. The Δ*dltA* mutant was used as control, since the DltA-E machinery is responsible for D-alanylation of TA, which introduces positively charged amines and thus prevents repulsive interactions between neighboring TA (*Perego et al., 1995*), similar to the effect of $Mg^{2+}$ incorporation in the cell wall. AFM confirmed that the absence of positive charges reduces cell wall rigidity in the Δ*dltA* control in regular TSB, as reported (*Saar-Dover et al., 2012*). In $Mg^{2+}$-enriched growth conditions, extracellular $Mg^{2+}$ binding complemented the cell wall rigidity defect in this mutant, as TA-coordinated $Mg^{2+}$ provided cell wall rigidity in the absence of the Dlt machinery. Our AFM measurements showed greater cell wall rigidity in $Mg^{2+}$-enriched growth conditions in the Δ*dltA* mutant (*Figure 3D*), comparable to the wild type strain. These experiments indicate that extracellular $Mg^{2+}$ is incorporated to cell wall TA to increase cell wall rigidity.

*Staphylococcus aureus* cell wall TA are essential for a response to extracellular $Mg^{2+}$, which increases BRcell subpopulation size and thus induces biofilm formation in TSBMg. In these conditions, cells treated with sublethal doses of tunicamycin, which inhibits TarO and thus teichoic acid synthesis at low concentrations (*Swoboda et al., 2010*; *Campbell et al., 2011*; *Nunomura et al., 2010*; *Swoboda et al., 2009*), did not respond to $Mg^{2+}$ and biofilm formation was inhibited (*Figure 3—figure supplement 1B*). Based on these findings, we genetically engineered *S. aureus* strains that down- and upregulate genes related to TA biosynthesis, such as *tagB* (*Figure 3—figure supplement 1*), verified *tagB* down- and upregulation in these strains by qRT-PCR (*Figure 3—figure supplement 1C*), confirmed that these strains show no significant defects in growth or peptidoglycan synthesis (*Figure 3—figure supplement 1D,E*) and tested their ability to form biofilms in TSBMg (*Figure 3—figure supplement 1F*). Strains with reduced *tagB* expression did not respond to $Mg^{2+}$ and thus did not develop biofilms (low-*tagB* strain) (*Figure 3E*). In contrast, strains with upregulated *tagB* became hypersensitive to extracellular $Mg^{2+}$ and produced more robust biofilms (high-*tagB*) even with $Mg^{2+}$ traces that are present in regular TSB medium (*Figure 3F*). We next tested whether the TA-mediated increase in cell wall rigidity downregulates *agr* bimodal behavior via $\sigma^B$ activation. To study this, we used qRT-PCR analysis to quantify the relative expression of the $\sigma^B$ target-gene *asp23* and staphyloxanthin quantification to determine $\sigma^B$ activation in low- and high-*tagB* strains (*Figure 3A* and *Figure 3—figure supplement 1G*). The low-*tagB* strain responded more weakly to extracellular $Mg^{2+}$ than the high-*tagB* strain, with limited $\sigma^B$ activation in both TSB and TSBMg conditions. In contrast, the high-*tagB* strain was hypersensitive to extracellular $Mg^{2+}$, with higher $\sigma^B$ activation than the other strains in TSB.

These results are consistent with our hypothesis that extracellular $Mg^{2+}$ stabilizes TA, increases cell wall rigidity and triggers the $\sigma^B$ inhibitory signal responsible for downregulating the *agr* bimodal switch. Once the *agr* switch is activated, variations in the concentration of these types of input signals affect switch activity and modulate the size of the two subpopulations. For instance, $Mg^{2+}$ in the colonization niche acts as a downregulatory signal, as it induces $\sigma^B$; activation of the *agr* switch becomes more difficult in these conditions and DRcell subpopulation size is reduced (*Figure 4A*). However, since this cue neither generates nor abolishes the *agr* positive feedback loop, but only modulates its activity, its effect would be restricted to varying the BRcell:DRcell ratio.

To substantiate this concept, we used quantitative analysis of fluorescence microscopy images and flow cytometry to monitor *S. aureus* cell differentiation in the presence of extracellular cues that influence the bimodal switch behavior (AIP excess and $\sigma^B$ activation). Purified AIP was added to $P_{psm\alpha}$-*yfp* or $P_{psm\beta}$-*yfp* reporter strain cultures at various concentrations above threshold concentration of ~10 µM usually found in cultures, which caused differentiation of a DRcell subpopulation that increased in parallel with AIP concentration but cell heterogeneity nonetheless remained detectable in cultures (*Figure 4B*). When we analyzed downregulation of the bimodal switch, WT cultures in $Mg^{2+}$-enriched growth conditions had a smaller DRcell subpopulation (*Figure 4A*), whereas the Δ*sigB* mutant in TSB and TSBMg media differentiated a larger DRcell subpopulation than the WT strain (*Figure 4C*). Nevertheless, in both cases BRcell and DRcell differentiation was detected in the Δ*sigB* mutant. Similarly, the low- and high-*tagB* strains, which are hypo- and hypersensitive to extracellular $Mg^{2+}$, showed larger and smaller DRcell subpopulations, respectively, in TSBMg (*Figure 4D*) although both subpopulations were detected.

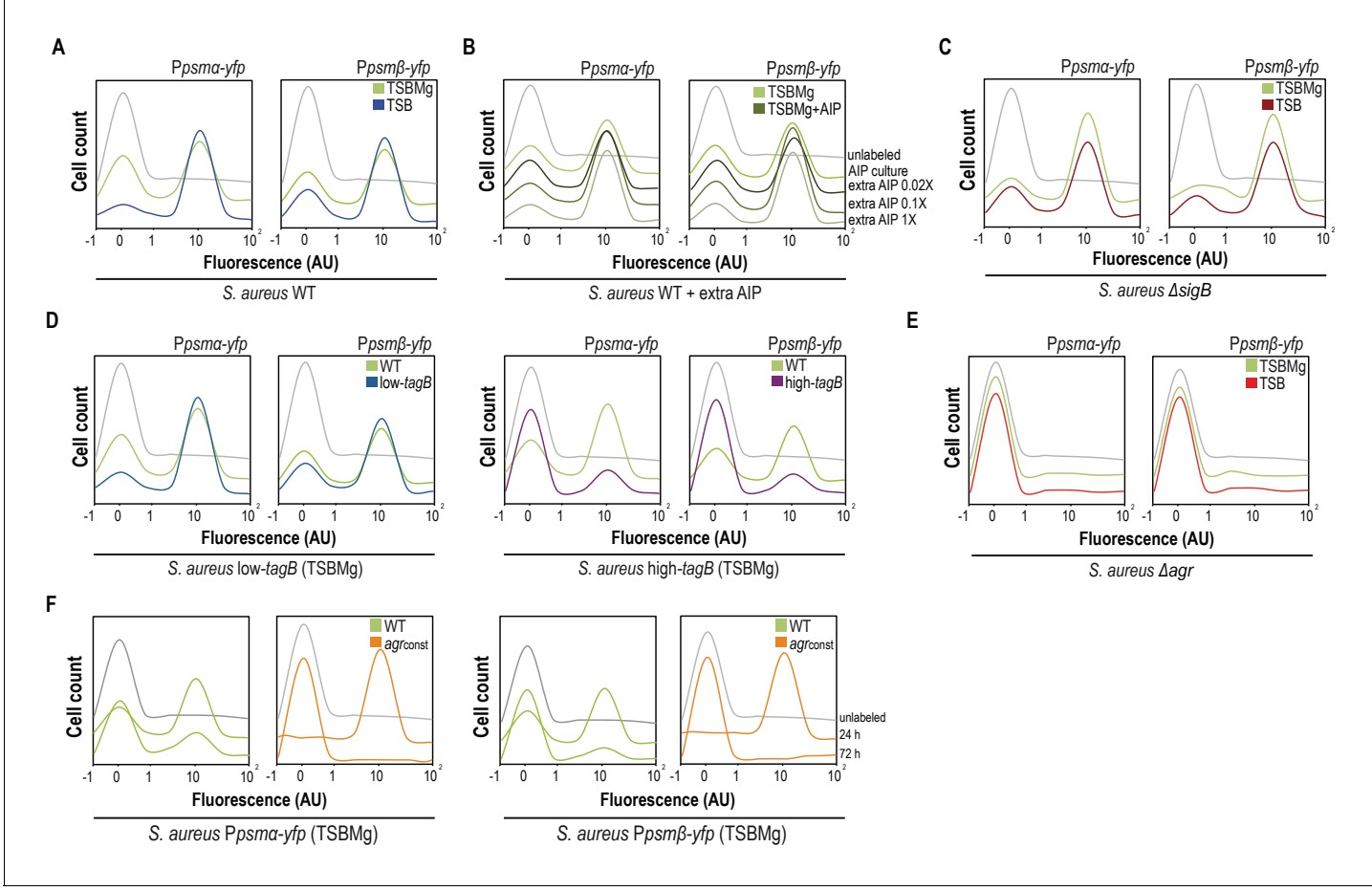

**Figure 4.** AIP and Mg$^{2+}$ modulate the BRcell:DRcell ratio in *S. aureus* communities. (**A**) Quantitative analysis of fluorescence microscopy images of *agr*-related promoters (P*psmα* and P*psmβ*) in TSBMg and TSB. We counted ~700 random cells from each of three independent microscopic fields from independent experiments (~2100 cells total for each strain). In the absence of extracellular Mg$^{2+}$, the proportion of DRcells increases in the staphylococcal community, in accordance with the role of Mg$^{2+}$ in repression of *agr* via σ$^{B}$. (**B**) Quantitative analyses of fluorescence microscopy images (n = 2100) of *agr*-related promoters in TSBMg with different concentrations of exogenous AIP1 (0.02x to 1x). Increasing AIP1 concentrations above threshold upregulates the *agr* bimodal switch and increases DRcell subpopulation size, although both on and off subpopulations are always detected ('AIP culture'=no exogenous AIP, equivalent to the 10 μM threshold concentration). (**C**) Quantitative analyses of fluorescence microscopy images (n = 2100) of *agr*-related promoters of *S. aureus*Δ*sigB* mutant in TSBMg and TSB. The Δ*sigB* mutant shows upregulation of the *agr* bimodal switch and differentiates a larger DRcell subpopulation. (**D**) Quantitative analysis of fluorescence microscopy images (n = 2100) of *agr*-related promoters in TSBMg and TSB, using engineered *S. aureus* strains that produce different TA levels (low-*tagB* and high-*tagB*). The differential sensitivity of these strains to extracellular Mg$^{2+}$ alters the proportion of DRcells within *S. aureus* aggregates. (**E**) Quantitative analyses of fluorescence microscopy images (n = 2100) of *agr*-related promoters of *S. aureus*Δ*agr* mutant in TSBMg and TSB. The Δ*agr* mutant lacks the bimodal switch that triggers cell differentiation. (**F**) Quantitative analysis of fluorescence microscopy images (n = 2100) of *agr*-related promoters in TSBMg, of *S. aureus* WT and a strain engineered to express the *agrBDCA* operon under the control of a constitutive promoter (*agr*_const); this disrupted the positive feedback loop, as the promoter that activates *agr* expression is no longer self-inducible. In the absence of a functional *agr* positive feedback loop, cell differentiation was not detected and P*psmα* and P*psmβ* reporter expression was homogeneous throughout the bacterial population.
DOI: https://doi.org/10.7554/eLife.28023.009

According to the information we obtained using the *agr* orthogonal system and to confirm that the only means by which to lead the system into unimodal gene expression is by disrupting the *agr* positive feedback loop, we monitored DRcell differentiation in the Δ*agr* strain. In this strain, expression of P$_{psmα}$-*yfp* or P$_{psmβ}$-*yfp* labeled reporters was not detected thus cell differentiation was inhibited in both TSB and TSBMg media (**Figure 4E**). Results from our orthogonal system pointed that the feedback loop activation mechanism relies on AgrA~P binding to the P2 promoter to turn on positive self-regulation of the *agr* operon, thus we monitored cell differentiation in a *S. aureus* strain in which P2 was replaced with a constitutive promoter (**Figure 4F**). This strain does not have an

active positive feedback loop, as the promoter that activates *agr* expression is no longer self-inducible. We monitored DRcell differentiation using P*psmα*-*yfp* or P*psmβ*-*yfp* reporter strains. In the absence of a functional *agr* feedback loop, we detected no cell differentiation and reporter expression was homogenous throughout the bacterial population. These results show that the *agr* positive feedback loop must be active to trigger cell differentiation, and that its activity is regulated by additional input cues that change the ratio of the specialized subpopulations.

## BRcells and DRcells organize spatially and are physiologically distinct cell types

Given that local Mg$^{2+}$ and AIP concentrations modulate *agr* switch activity, we explored spatial organization of cell types during colony development, as reported for other bacteria (*Yarwood et al., 2007*). We developed a mathematical model that considers these factors in the context of nutrient availability and bacterial growth, and delineates growth of multicellular aggregates as a non-linear reaction-diffusion equation system (*Figure 5A* and *Figure 5—figure supplement 1A*) (*Hense et al., 2012*). Based on the morphological traits of the multicellular aggregates in different genetic backgrounds (*Figure 5—figure supplement 1B–G*), the model predicted that as an aggregate grows and expands, nutrients become limited in the older, central biofilm region, which has higher AIP levels and slow-dividing cells, which increases representation of the DRcell subpopulation. We sectioned mature aggregates into concentric, morphologically distinct regions and analyzed DRcell and BRcell subpopulation size by flow cytometry (*Figure 5B* and *Figure 5—figure supplement 2A*). In accordance with the mathematical predictions, the most peripheral region had a larger proportion of BRcells and a smaller proportion of DRcells. DRcells were enriched in the aggregate center. We combined cryosectioning and confocal microscopy to determine subpopulation size and location within the inner zones of thick biofilms (*Figure 5C* and *Figure 5—figure supplement 2B*). BRcells were highly represented in regions near the aggregate outer edge, where nutrient concentration is high (*Cramton et al., 2001*), whereas DRcell representation was more prominent in the biofilm inner region, further from the nutrient source. These experiments showed enrichment of BRcells in newer and of DRcells in older biofilm regions, suggesting that the staphylococcal cells respond differently to local input signal concentrations, and differentiate distinct DRcell:BRcell ratios in different biofilm regions.

To study the potential physiological specialization of BRcell and DRcell types beyond the differential expression of *agr*-regulated reporters, we determined their transcriptional profiles using Illumina RNA-sequencing after enrichment by fluorescence-activated cell sorting (*Figure 6* and *Figure 6—figure supplement 1*). We grew separate mature aggregates of the strains labeled with the P*psmα*-*yfp* and P*ica*-*yfp* reporter fusions to identify the DRcell and BRcell subpopulations, respectively. Fluorescent cells from mature aggregates were sorted from the rest of the cell population; both fluorescent cells (enriched) and whole cell community (non-enriched) were collected simultaneously in separate samples. Genome-wide analysis showed similar genetic landscapes for DRcell and BRcell subpopulations, indicating that cell differentiation is not the result of accumulated mutations (*Figure 6—source data 1*). RNA was purified from each sample prior to Illumina Hi-seq RNA sequencing. The total number of reads allowed mapping of ~96% to the *S. aureus* genome (*Figure 6—figure supplement 2A–B*). Comparison of the normalized gene expression profiles (*Figure 6A*, *Figure 6—figure supplement 2C* and *Figure 6—source datas 2–4*) and qRT-PCR validation (*Figure 6—figure supplement 2D–E*) showed marked differences between the enriched subpopulation and its non-enriched counterpart, which suggested that a specific physiological state could be attributed to each particular cell type.

BRcells had a large number of upregulated genes, including *sigB* sigma factor, and of many biofilm-related genes as well as genes related to peptidoglycan turnover, cell division and DNA replication. In addition, 49 tRNAs were upregulated, which indicates the higher metabolic activity of BRcells and their physiological predisposition to proliferation (*Figure 6B* and *Figure 6—figure supplement 2D*). DRcells showed a smaller number of upregulated genes, which we attributed to lower gene expression activity potentially due to the lower physiological activity of this cell type. Among the few upregulated genes detected, we found a notable number related to toxin secretion and host invasion, such as the type-VII secretion system (*Burts et al., 2005*), as well as genes related to protecting bacterial cells from the immune system, such as the *hssRS-htrAB* hemin detoxification system (*Stauff et al., 2007*). We also detected upregulation of multi-drug efflux pumps that confer

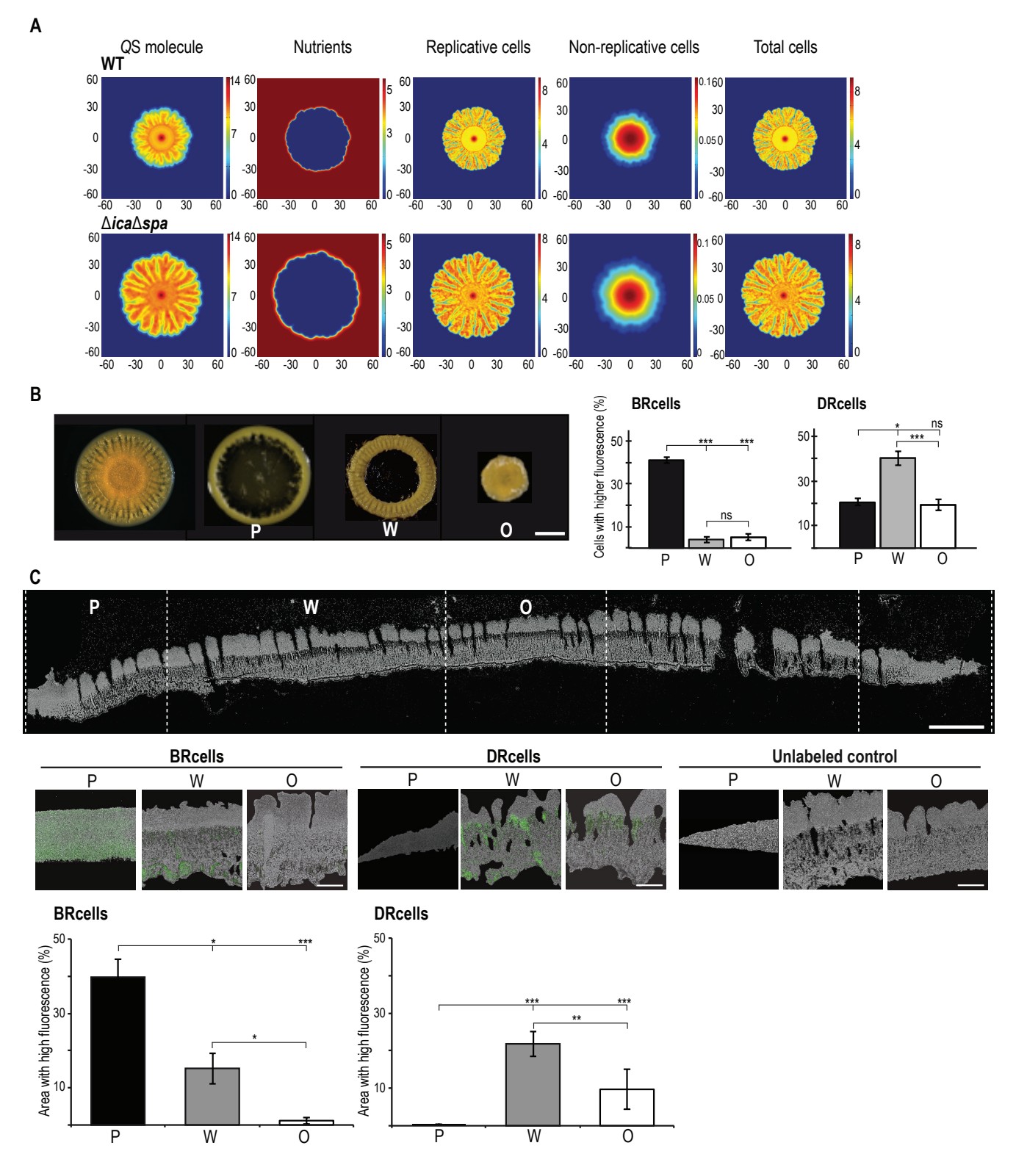

**Figure 5.** Collective behavior of BRcells and DRcells in vitro. (**A**) Mathematical modeling of WT and a matrix-deficient mutant (Δ*ica*Δ*spa*) *S. aureus* growth in multicellular aggregates. Local concentrations of factors that affect AIP activity are represented on a color scale. (**B**) Left, dissection of a 5 day multicellular aggregate into three distinct, concentric morphological regions, a peripheral region (**P**), a surrounding wrinkled area (**W**) and a central older region or origin (**O**); each one can be separated from the aggregate and processed independently. Bar = 5 mm. Right, quantitative analysis of

*Figure 5 continued on next page*

*Figure 5 continued*

flow cytometry data for BRcells and DRcells in concentric regions of a mature aggregate ($n$ = 50,000). (C) Top, longitudinal-transverse cryosection of a mature *S. aureus* aggregate. Bar = 500 μm. Center, spatial distribution of BRcells and DRcells within the aggregate. Bar = 100 μm. Bottom, quantitative estimate of BRcells (*ica* and *spa* reporters) and DRcells (P*psmα* and P*psmβ*reporters) fluorescent area over the total thin section aggregate area in representative images (see Material and methods for quantification details). Statistical significance was measured using one-way ANOVA and Tukey's test for multiple comparisons. *p<0.05, **p<0.01, ***p<0.001; ns, no significant differences. Data shown as mean ±SD of three independent experiments ($n$ = 3).

DOI: https://doi.org/10.7554/eLife.28023.010

The following figure supplements are available for figure 5:

**Figure supplement 1.** *Staphylococcus aureus* develops architecturally complex multicellular aggregates in Magnesium-supplemented TSB medium (TSBMg).

DOI: https://doi.org/10.7554/eLife.28023.011

**Figure supplement 2.** Spatial distribution of *BRcells* and *DRcells* in vitro.

DOI: https://doi.org/10.7554/eLife.28023.012

resistance to diverse antimicrobials, and of regulators such as *graR* and *arsR*, which positively control gene-related cell-wall antibiotic resistance and metal ion stress (*Figure 6B* and *Figure 6—figure supplement 2E*). In contrast, the expression of selected housekeeping genes in aggregates growing in TSBMg and TSB showed no differences (*Figure 6—figure supplement 2F*), suggesting that extracellular $Mg^{2+}$ specifically influences the *agr* bimodal switch pathway rather than causing a global deregulation of gene expression. These results suggest that the DRcell subpopulation has lower metabolic activity than BRcell and is predisposed to resist different types of antimicrobials.

## BR and DR cell fates arise during *in vivo* infections

An important question arises from the physiological differences detected between BRcells and DRcells *in vitro*, concerning the role and impact of these cell types during progression of *S. aureus* infections. We found that extracellular $Mg^{2+}$ is an input signal that regulates *agr* activity and modulates the sizes of specialized cell subpopulations *in vitro*, and hypothesized that a similar correlation could be found *in vivo*, that is, that different tissue $Mg^{2+}$ concentrations would lead to distinct subpopulation ratios and distinct infection outcomes. To address this question, we developed an infection model (*Koch et al., 2014*) in which mice were intravenously infected with $10^7$ colony forming units (CFU) of cells labeled with $P_{ica}$-*yfp* (BRcells) or $P_{psmα}$-*yfp* (DRcells) reporters. Infections were allowed to progress for four days, after which mice were sacrificed and the infected organs collected (*Figure 7* and *Figure 7—figure supplement 1*).

The *in vitro* results correlated with the *in vivo* experiments; bacteria proliferated more efficiently in $Mg^{2+}$-rich organs such as kidney. Infected kidneys showed a bacterial load of $10^{10}$ CFU/g of tissue (*Figure 7—figure supplement 1A*), and histological preparations of these organs showed large bacterial aggregates surrounded by immune cell infiltrates (*Figure 7A*, *Figure 7—figure supplement 1B* and *Figure 7—figure supplement 3*), indicative of long-term colonization during septicemia (*Prabhakara et al., 2011*). Confocal microscopy analyses showed approximately three-fold more BRcells than DRcells in kidney aggregates (*Figure 7C Figure 7—figure supplement 2*), similar to levels detected in in vitro experiments; this was consistent with reports that kidneys are $Mg^{2+}$ reservoirs in the body (*Günther, 2011*; *Jahnen-Dechent and Ketteler, 2012*), and that 82% of patients with urinary catheterization develop long-term *S. aureus* infections (*Muder et al., 2006*).

On the other hand, infected hearts showed a bacterial load of $10^7$ CFU/g of tissue (*Figure 7—figure supplement 1A*), which suggested that *S. aureus* cells that colonized heart tissues proliferated less actively than those in kidney. Infected hearts had a larger DRcell subpopulation, consistent with the lower metabolic activity, the lower proliferation rate of these cells in vitro and the lower $Mg^{2+}$ concentration typically found in heart tissue (*Günther, 2011*; *Jahnen-Dechent and Ketteler, 2012*). Histological preparations of infected hearts revealed deposits of disperse cells with no immune cell infiltrates (*Figure 7B*, *Figure 7—figure supplement 2* and *Figure 7—figure supplement 3*), which is indicative of acute bacteremia (*McAdow et al., 2011*). Confocal microscopy analysis showed that as much as 60% of the total heart tissue-colonizing bacterial population consists of DRcells (*Figure 7D* and *Figure 7—figure supplement 2*), as observed in *in vitro* experiments.

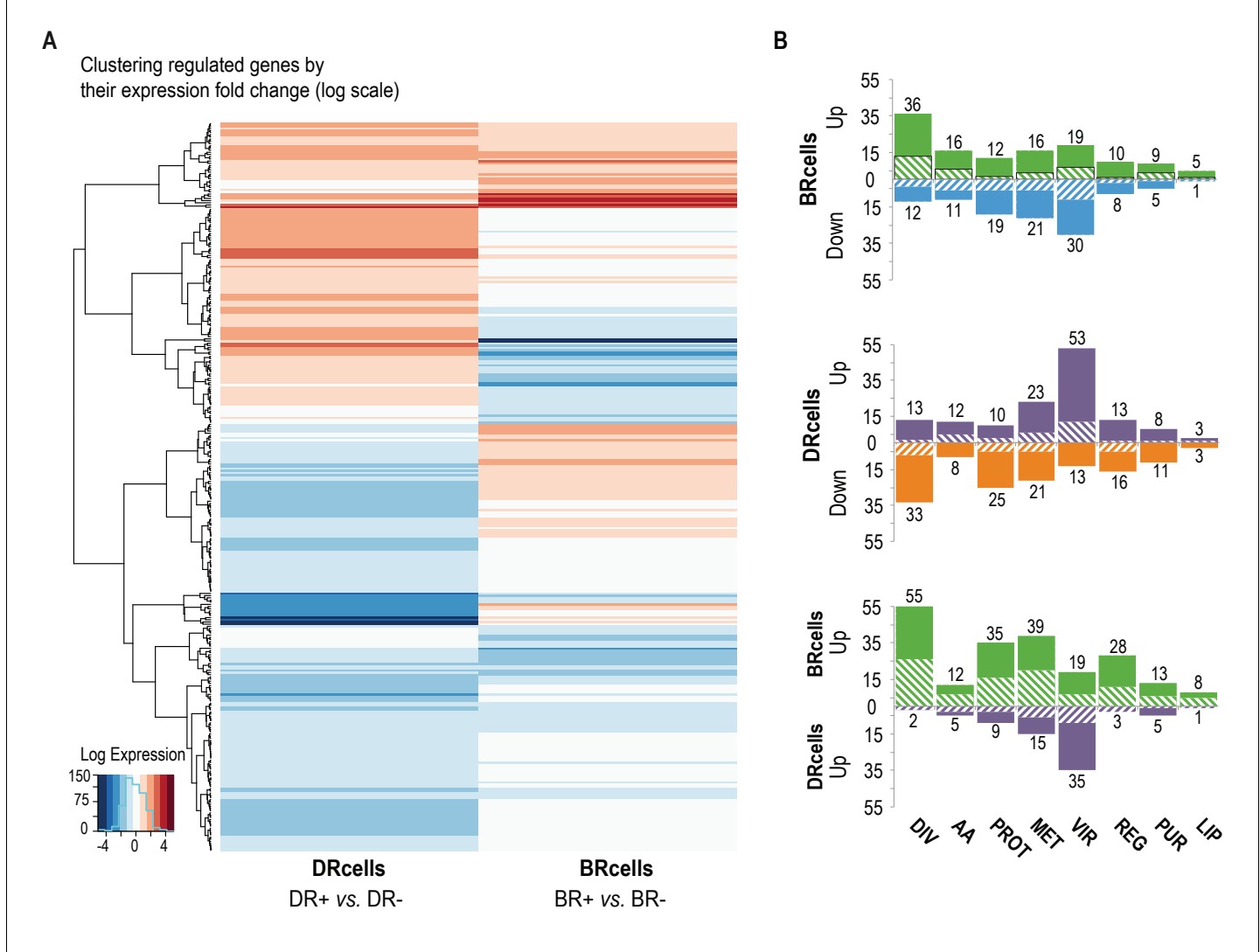

**Figure 6.** BRcell and DRcell subpopulations have different gene expression profiles. (**A**) Unsupervised hierarchical clustering of commonly expressed genes differentially regulated in at least one of the libraries shows a specific, divergent expression profile for BRcells and DRcells. Color scales represent log₂ fold-changes for differential expression. Clustering was carried out on the regulated genes (minimum fold-change 2) with Ward hierarchical biclustering using the heatmap.2 command in the ggplots package of the R programming language on Euclidean distances. This approach successfully grouped the common genes, which are upregulated (orange) in both sets of libraries, and far from the cluster of downregulated genes (blue). The third set of genes was identified based on this clustering (in the center of the heatmap), which showed library-specific phenotypes (upregulated in one library and downregulated in the other). (**B**) Classification of the differentially expressed genes using TIGRfam, SEED and Gene Ontology (GO) functional categories, followed by manual curation. For each category, solid and dashed columns represent the number of regulated genes from DESeq analysis using the raw read threshold of >8 and >25, respectively. *DIV*, DNA replication, cell envelope and cell division; *AA*, amino acid synthesis; *PROT*, protein synthesis and processing; *MET*, energy and intermediary metabolism; *VIR*, virulence, binding and transport; *REG*, regulation, transcription and signal transduction; *PUR*, purines, pyrimidines, nucleotides and nucleosides; *LIP*, lipid metabolism.
DOI: https://doi.org/10.7554/eLife.28023.013

The following source data and figure supplements are available for figure 6:

**Source data 1.** Genome-wide analysis of DRcell and BRcell sorted subpopulations.
DOI: https://doi.org/10.7554/eLife.28023.016

**Source data 2.** Gene quantification and differential expression analysis of *BRcells* and *DRcells*.
DOI: https://doi.org/10.7554/eLife.28023.017

**Source data 3.** Functional classification of annotated genes.
DOI: https://doi.org/10.7554/eLife.28023.018

*Figure 6 continued on next page*

*Figure 6 continued*

**Source data 4.** Hypergeometric analysis for library comparison comprising the log2fold values for differentially expressed genes that were shown to be expressed by DESeq analysis comparison of *DRcells+* against *BRcells+* sample sets.
DOI: https://doi.org/10.7554/eLife.28023.019

**Figure supplement 1.** Experimental workflow to sort BRcells and DRcells using Fluorescence Activated Cell Sorting (FACS) to analyze and compare their transcriptomic profile.
DOI: https://doi.org/10.7554/eLife.28023.014

**Figure supplement 2.** Read alignment statistics, transcriptomic profile of BRcells and DRcells and differential distribution of cell types in distinct infected organs.
DOI: https://doi.org/10.7554/eLife.28023.015

To further correlate the presence of extracellular $Mg^{2+}$ with infection outcome, we performed infection studies using the suite of low- and high-*tagB* strains (*Figure 8A*). Kidneys showed a reduction in bacterial load when infected with the $Mg^{2+}$-hyposensitive low-*tagB* strain, although this strain colonized heart tissues more efficiently than *S. aureus* WT. qRT-PCR analyses verified that these differences were associated with upregulation of key genes whose expression is restricted to DRcells (*agrA*, *agrB* and *psmα/β*), which suggests that infections with a low-*tagB* strain had marked representation of DRcells (*Figure 8B–C*). In contrast, the $Mg^{2+}$-hypersensitive high-*tagB* strain was able to infect kidneys more efficiently than the WT strain, concomitant with a reduced infection of heart tissues. qRT-PCR analyses showed higher expression of genes related to BRcells (*icaA*, *icaB* and *spa*), which suggested that the high-*tagB* strain differentiated a larger subpopulation of BRcells. We generated a new strain derived from the high-*tagB* strain that also lacks σ$^B$. Higher TA content in this strain increases cell wall rigidity in response to $Mg^{2+}$, but the lack of σ$^B$ should prevent activation of biofilm formation via downregulation of *agr*. The infection pattern of this Δ*sigB* high-*tagB* strain resembled the low-*tagB* pattern in all the organs analyzed. Kidneys infected with this strain thus showed a reduction in bacterial load, while heart tissues were colonized more efficiently. qPCR analyses demonstrated higher expression of genes related to DRcells in infected kidneys and heart tissues, pointing to a larger number of DRcells in the infection of these strains. In addition, infected livers, which had a moderate of $Mg^{2+}$ concentration (*Günther, 2011*; *Jahnen-Dechent and Ketteler, 2012*), also showed comparable bacterial loads between low-*tagB*, high-*tagB* and Δ*sigB* high-*tagB* strains.

## Discussion

The bimodal behavior of the *agr* system results in the differentiation of two genetically identical, cell types that specialize in biofilm- or dispersal-associated lifestyles in *S. aureus* communities. These cell types were detected in in vitro cultures and during in vivo infections. The expression of specific markers by *BRcells* and *DRcells* is due to activation of the *agr* bimodal switch, which requires the production of the activating signal AIP above a certain threshold, similar to other quorum sensing systems, such as the heterogeneous expression of natural competence in cultures of *B. subtilis* in response ComX signal (*Maamar and Dubnau, 2005*; *Smits et al., 2005*) or quorum sensing activation of bioluminescence in a subpopulation of cells in *Vibrio harveyi* (*Anetzberger et al., 2009*). A growing number of bimodal switches are being found in QS pathways, and cell differentiation in response to a QS signal is becoming an established concept in microbiology.

The molecular mechanism of *agr* bimodal behavior is based on a sequential activation of the two adjacent divergent promoters P2 and P3. The P2 promoter triggers positive self-regulation of the *agr* operon (by activating the *agr* positive feedback loop) and the P3 promoter induces the *agr* regulon responsible for activation of virulence genes (P3 promoter activation is necessary for DRcell specialization). The differential affinity of AgrA~P for P2 and P3 promoters is crucial for the sequential promoter activation and thus for *agr* bimodal switch activation. AgrA~P binds P2 with greater affinity than the P3 promoter. The P2 promoter thus activates and triggers the feedback loop at lower AgrA~P concentrations and only in a given subpopulation (*agr*-on cells). Activation of this feedback loop produces high AgrA~P levels in this subpopulation, which licenses them to trigger the less-sensitive P3 promoter and induce the *agr* regulon, leading the *agr*-on cells to specialize in dispersion and virulence and become DRcells. In contrast, the cell subpopulation that expresses P2 below the

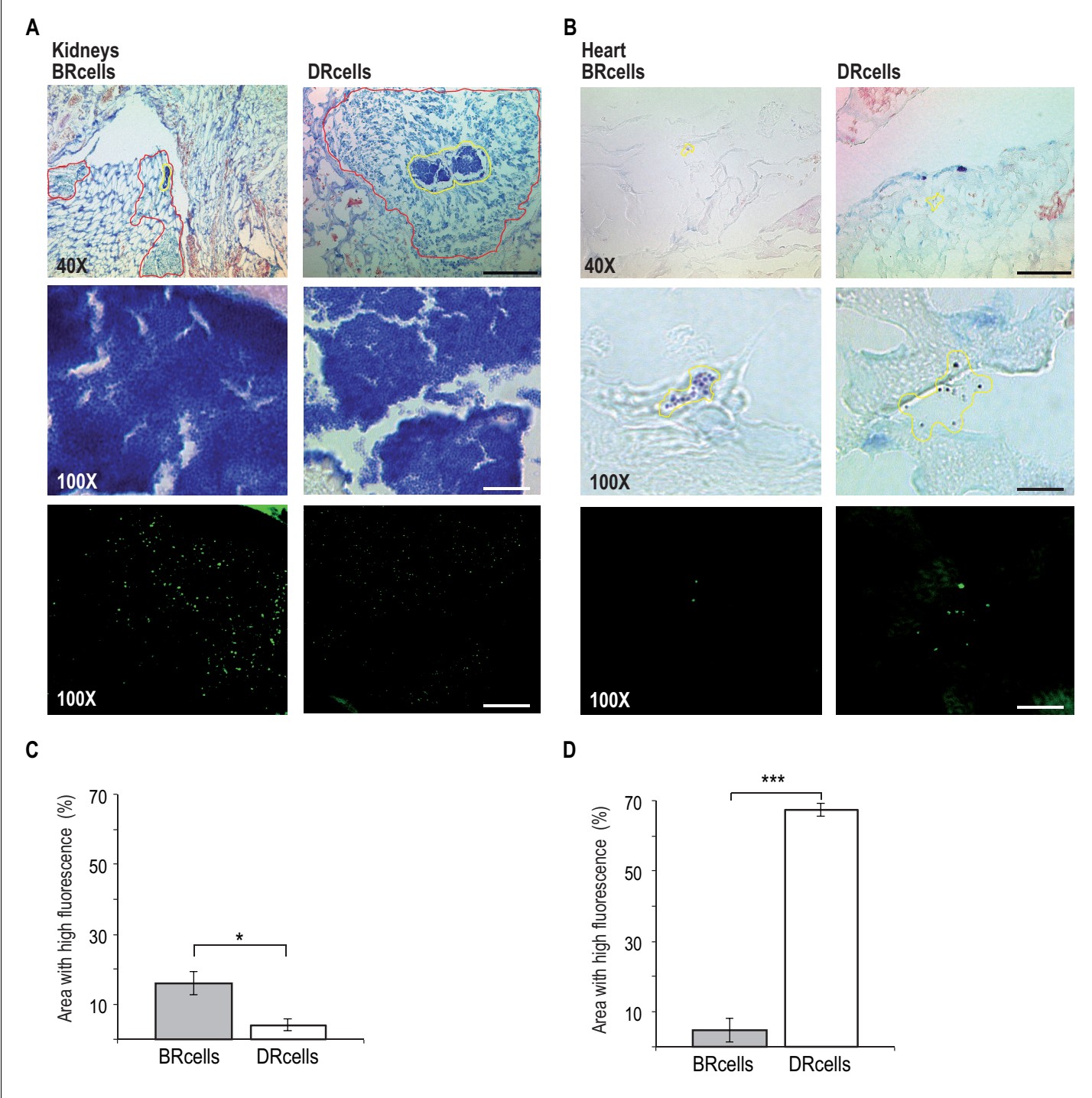

**Figure 7.** BRcells are more represented in infected kidneys and DRcells are more represented in infected hearts. Histological preparations of infected kidneys (**A**) or hearts (**B**) stained with Giemsa solution and visualized using light microscopy. Upper row, 40X magnification of the preparation. Bar = 50 μm. The area delineated in red corresponds to the immune cell infiltrates that surround bacterial aggregates in infected kidneys. The area delineated in yellow corresponds to the bacterial aggregates surrounded by immune cell infiltrates in infected kidneys or, to dispersed *S. aureus* cells in the case of infected hearts. This area is magnified at 100X in central row. Central row, compact aggregates of *S. aureus* cells can be seen in dark blue. Bar = 20 μm. Bottom row, confocal fluorescence microscopy images of the bacterial populations imaged in row 3. Right, monitoring of BRcell subpopulation using a P$_{ica}$-*yfp S. aureus* labeled strain. Left, monitoring of DRcell subpopulation using a P$_{psmα}$-*yfp S. aureus* labeled strain. Magnification, 100X. The fluorescence signal is shown in green. Bar = 20 μm. (**C and D**) Quantitative estimate of the relative fluorescent signal is shown as a percentage of the

*Figure 7 continued on next page*

*Figure 7 continued*

fluorescent area over the total bacterial aggregate area in the images. Statistical significance was measured by an unpaired, two-tailed Student's t-test. *p<0.05. Data shown as mean ± SD of three independent measurements (*n* = 3) each one obtained from different infected organs.

DOI: https://doi.org/10.7554/eLife.28023.020

The following figure supplements are available for figure 7:

**Figure supplement 1.** Bacterial loads in Mg$^{2+}$-enriched and Mg$^{2+}$-depleted organs.

DOI: https://doi.org/10.7554/eLife.28023.021

**Figure supplement 2.** BRcells are more represented in infected kidneys and DRcells are more represented in infected hearts.

DOI: https://doi.org/10.7554/eLife.28023.022

**Figure supplement 3.** BRcells are more represented in infected kidneys and DRcells are more represented in infected hearts.

DOI: https://doi.org/10.7554/eLife.28023.023

threshold cannot activate the *agr* positive feedback loop (*agr*-off cells), and thus do not produce sufficient AgrA~P to induce P3 promoter expression. In these cells, genes normally repressed by

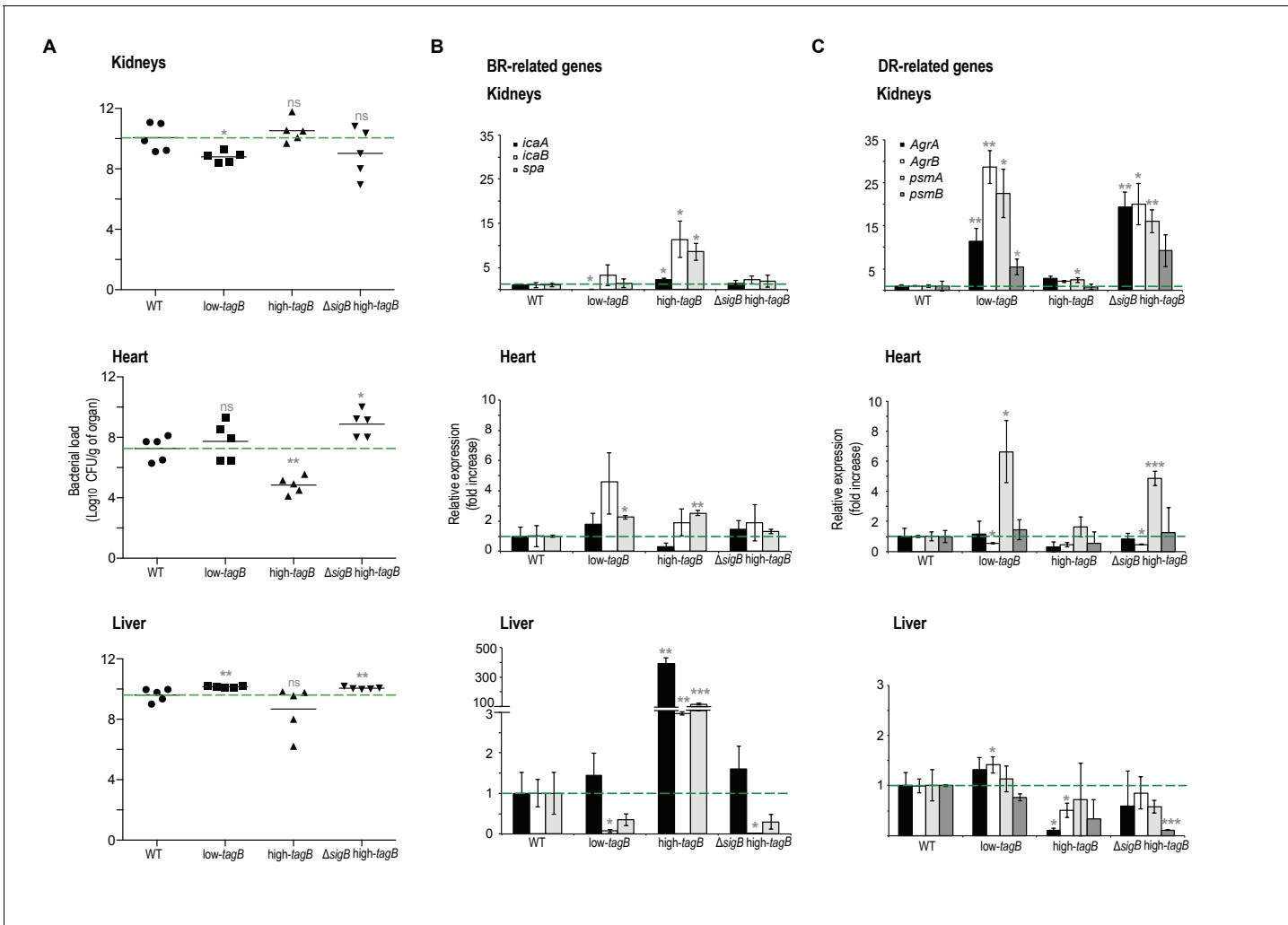

**Figure 8.** Low- and high-*tagB* strains display different infection patterns. (A) Bacterial loads on different genetic backgrounds in kidney, heart and liver of infected mice. (B, C) qRT-PCR analysis of BRcell- (B) and DRcell- (C) related genes in kidney, heart and liver of mice infected with *S. aureus* strains of different genetic backgrounds. Data shown as mean ± SD of five independent animals (*n* = 5) (A) and three independent experiments (*n* = 3) (B, C). Statistical significance was measured by multiple comparison analysis using the Mann-Whitney test (A) and unpaired two-tailed Student's t-test (B, C). *p<0.05, **p<0.01, ***p<0.001; ns, not significant differences.

DOI: https://doi.org/10.7554/eLife.28023.024

AgrA~P are upregulated, including biofilm-related genes, which licenses them to differentiate as biofilm-producing cells thus to become BRcells.

Once the *agr* bimodal switch is activated and BRcells and DRcells differentiate, subpopulation size is modulated by other extracellular cues that affect bimodal switch activity. We report that extracellular $Mg^{2+}$ is incorporated into the bacterial cell wall by binding teichoic acids and increases cell wall rigidity, provoking activation of *sigB* expression. As $\sigma^B$ downregulates the *agr* bimodal switch, its activation above the threshold becomes more difficult, leading to differentiation of a smaller DRcell and a larger BRcell subpopulation, which in turn facilitates biofilm formation. In contrast, increasing AIP above the threshold concentration facilitates activation of the *agr* bimodal switch, leading to differentiation of a larger DRcell and a smaller BRcell subpopulation, which in turn facilitates dispersion and acute infection.

The activity of these *agr* input cues neither generates nor abolishes the *agr* positive feedback loop; *agr* bimodal switch activity is only modulated, which regulates the size of the two subpopulations. DRcell and BRcell subpopulations can therefore be detected in the presence of input cues; in our in vitro and in vivo assays, only their ratio differed in the overall bacterial community, thus varying infection outcome in distinct scenarios. In mouse models, we found that in vivo infections developed a larger DRcell subpopulation in tissues with low $Mg^{2+}$ levels (cardiac tissue), in which the microbial population became dispersed. In contrast, BRcells were more prevalent in tissues with high $Mg^{2+}$ concentrations (bone, kidney) and bacteria organized in microbial aggregates characteristic of biofilm-associated infections. The results recapitulate clinical studies in which a marked number of biofilm-associated persistent infections develop in urinary tract and bone, usually following surgery or catheterization (*Muder et al., 2006*; *Brady et al., 2008*; *Flores-Mireles et al., 2015*; *Idelevich et al., 2016*). These data help to understand how nosocomial pathogens such as *S. aureus* can simultaneously cause dissimilar types of infections in distinct organs, and show that cell differentiation in nosocomial pathogens is particularly relevant for adaptation to different host tissues.

The molecular mechanism whereby extracellular $Mg^{2+}$ downregulates the *agr* bimodal switch relies on the capacity of magnesium to bind TA to increase cell wall rigidity (*Heptinstall et al., 1970*; *Hughes et al., 1971*), in turn causing *agr* downregulation via activation of the repressor $\sigma^B$. *Staphylococcus aureus* cells with reduced cell wall TA adhere poorly, have poor biofilm formation ability (*Vergara-Irigaray et al., 2008*), and do not colonize nasal (*Weidenmaier et al., 2004*) or kidney-derived endothelial tissues (*Weidenmaier et al., 2005a*). An effect similar to that of $Mg^{2+}$ incorporation into the cell wall is caused by D-alanine esterification of TA (*Lambert et al., 1975a*; *Lambert et al., 1975b*). The Dlt protein machinery decorates TA with D-alanine esters to reduce repulsive interactions between TA negative charges, which also increases cell wall rigidity (*Perego et al., 1995*). Previous reports showed that Dlt activity is important for biofilm formation in *S. aureus* (*Gross et al., 2001*) and for developing biofilm-associated infections in vivo in animal models (*Weidenmaier et al., 2005b*). Our results are consistent with these findings and suggest that, given the intricacy of *agr* bimodal switch regulatory control, many extracellular cues probably contribute to the outcome of *S. aureus* infections. Extracellular $Mg^{2+}$ and hence, increased cell wall rigidity are probably important cues for triggering biofilm-associated infections.

Here we show that infections generated by clonal populations of bacteria can bifurcate into distinct, specialized cell types that localize physically in different colonization tissues in the course of an infection. The prevalence of physiologically distinct bacterial cells in different organs could offers bacteria an adaptive strategy that increases their chance of evading the immune system evasion during infection, or provides a bet-hedging strategy that increases the possibility of survival during antimicrobial therapy (*Beaumont et al., 2009*). Understanding cell differentiation in bacteria is central to designing new anti-infective strategies that target specific cell subpopulations, particularly to pathogens like *Staphylococcus aureus*, considered endemic in hospitals and which has an approximately 20% mortality rate (*Klevens et al., 2007*).

## Materials and methods

### Strains, media and culture conditions

A complete list of strains used in this study is shown in *Supplementary file 1*. The laboratory *S. aureus* strain RN4220 (*Kornblum et al., 1990*) was used for cloning purposes. The strain *Escherichia*

coli DH5α was used for propagating plasmids and genetic constructs in laboratory conditions. *B. subtilis* and *E. coli* strains were regularly grown in LB medium. When required, selective media were prepared in LB agar using antibiotics at the following final concentrations: ampicillin 100 μg/ml, kanamycin 10 μg/ml, chloramphenicol 5 μg/ml, tetracycline 10 μg/ml, and erythromycin 2 μg/ml. *S. aureus* strains were routinely propagated in liquid TSB medium incubated with shaking (220 rpm) at 37°C for 16 hr. When required, selective media were prepared in TSB using antibiotics at the following final concentrations: kanamycin 10 μg/ml, chloramphenicol 10 μg/ml, tetracycline 10 μg/ml, erythromycin 2 μg/ml, neomycin at 75 μg/ml, Spectinomycin at 300 μg/ml and Lincomycin at 25 μg/ml.

For *S. aureus* aggregates in TSBMg, 4 μl of an overnight liquid culture was spotted in TSBMg (TSB medium supplemented with $MgCl_2$ 100 mM) (*Koch et al., 2014*) and dried in a sterile culture cabin. Plates were allowed to grow for five days at 37°C (*Koch et al., 2014*). For *S. aureus* traditional biofilms in liquid TSBMg and TSB, we followed the protocol proposed by O'Toole and Kolter (*O'Toole and Kolter, 1998a*). Briefly, an overnight liquid culture was inoculated 1:100 in fresh TSB media and after 6 hr at 37°C with shaking at 220 rpm, the $OD_{600}$ nm was measured and normalized to a final $OD_{600}$ 0.05. From the normalized cultures, 5 μl was inoculated in 995 μl of TSB or TSBMg in a 24-well microtiter plate (Thermo) (RRID:SCR_008452) and incubated for 24 or 48 hr at 37°C without shaking. Biofilm formation was measured as follows: media was discarded by aspiration; wells were washed twice with PBS and allowed to dry for 45 min at 65°C. Then, 500 μl of a solution of Crystal Violet at 0.1% was added and to stain the organic material associated with the well for 5 min. After staining, wells were washed three times with deionized water. For quantitative analyses, Crystal Violet was solubilized using 500 μl of acetic acid at 33%. The solubilized dye was diluted 1:100 in deionized water and transferred to 96-well microtiter plates (Thermo). The absorbance was determined at 595 nm using an InfiniteF200 Pro microtiter plate reader (Tecan). Background was corrected by subtracting the absorbance values of non-bacteria inoculated wells. Specific growth conditions are presented in figure legends.

For the experiments using the synthetic orthogonal *agr* model generated in *B. subtilis* wild type and Δ*sigB* mutant, cells were incubated in LB medium at 220 rpm at 37°C until cultures reached an $OD_{600}$ nm = 0.5. After incubation, 50 μl of the culture was added to 50 ml of chemically-defined MSgg growing medium (*Branda et al., 2001*) and allowed to grow for 4 hr at 37°C at 220 rpm. After incubation, 50 μl of culture was used to inoculate 50 ml of fresh LB and allowed to grow for 12 hr at 37°C with constant shaking. Addition of AIP to the culture defined the initiation of the experiment (time = 0 hr). Samples were taken at 0 hr, 2 hr, 4 hr, 6 hr, 8 hr, 10 and 12 hr.

## Strain generation

To generate the *S. aureus* strain Newman Δ*ica*, Δ*psmα*, Δ*psmβ* and Δ*dltA-E* mutants, 500 bp flanking each gene and the respective antibiotic cassettes were PCR amplified and the fragments were subsequently joined together using a long-flanking homology PCR (LFH-PCR). The resulting fragments were cloned into pMAD plasmid (*Arnaud et al., 2004*) and transformed into the laboratory strain *S. aureus* RN4220. To transfer the mutations from RN4220 to Newman and to generate the double mutant strains, φ11 phage lysates were generated from RN4220 mutants to infect Newman, NewHG and USA300 LAC* (*Rudin et al., 1974*). Clones resistant to the respective antibiotic were further verified to carry the mutation using PCR. Positive clones were validated to carry the mutation using Sanger sequencing.

To generate the *S. aureus* strains single-labeled with $P_{ica}$-*yfp*, $P_{spa}$-*yfp*, $P_{psmα}$-*yfp*, $P_{psmα}$-*mars*, $P_{psmβ}$-*yfp*, $P_{dnaA}$-*yfp* and double-labeled with $P_{spa}$-*yfp* $P_{ica}$-*mars*, $P_{psmβ}$-*yfp* $P_{psmα}$-*mars*, $P_{psmα}$-*yfp* $P_{ica}$-*mars*, $P_{spa}$-*yfp* $P_{psmα}$-*mars*, $P_{ica}$-*yfp* $P_{clfA}$-*mars*, $P_{ica}$-*yfp* $P_{isdA}$-*mars*, $P_{spa}$-*yfp* $P_{clfA}$-*mars* and $P_{spa}$-*yfp* $P_{isdA}$-*mars* transcriptional fusions, the respective promoter region comprising 200 to 500 bp upstream of the start codon was fused to the *yfp* reporter-gene using the plasmid pKM003 or to the *rfp* (mars) reporter-gene using the plasmid pKMmars. These fusions were subcloned into the plasmids pAmy and pLac (*Yepes et al., 2014*). The plasmids were integrated into the neutral *amy* and *lac* loci of *S. aureus* chromosome to ensure a uniform and chromosome-equivalent copy number of the reporters in all the cells within the microbial community. The integration of reporters in *amy* and *lac* neutral loci occurs in a two-step recombination process, as described in (*Yepes et al., 2014*). Briefly, integration of the plasmid into the chromosome of *S. aureus* occurs via a single recombination event. This first recombination occurs by growing the plasmid-carrying strain overnight at 30°C,

plating serial dilutions onto selective media (erythromycin 2 µg/ml and X-Gal 100 µg/ml) and incubating the plates at 44°C. This is a temperature-sensitive plasmid, which does not allow plasmid replication at higher temperatures (*Arnaud et al., 2004*). Therefore, incubation at 44°C allows only the strains that incorporate the plasmid into the chromosome to grow. The genetic constructs obtained from the first recombination process in the strain RN4220 were transferred to strain Newman by φ11 phage transduction (*Rudin et al., 1974*). Once the constructs were transferred to the recipient strain (Newman and USA300 LAC* strains), we forced a second recombination process to leave only the reporter in the integration locus. To do this, a culture of a light-blue colony was incubated overnight at 30°C in the absence of erythromycin, plating serial dilutions onto selective (erythromycin and X-Gal) media and then incubating the plates at 44°C. After 48 hr of incubation at 44°C, the light-blue colonies still carrying the plasmid in the chromosome were discarded. A four-step process of screening, including fluorescence, antibiotic susceptibility, PCR and Sanger sequencing, was designed to validate whether the white colonies carried the corresponding insertion in the neutral loci. The *S. aureus* strains *tagB*-lower (NWMN_0187), *tagG*-lower (NWMN_1763) and *tagH*-lower (NWMN_1763) were obtained by phage transduction using as donor strain the respective mutants deposited in the transposon-mapped mutant collection (*Bae et al., 2004*). Clones were verified using PCR and Sanger sequencing. The *S. aureus* strain that overexpresses the *tagB* gene (*tagB*-higher) (NWMN_0187) or the *agrBCDA* operon (NWMN_1943 to NWMN_1946) were obtained by cloning the complete ORF into the replicative plasmid pJL74 (*Klijn et al., 2006*). The *sarA* P1 promoter and the RBS of *sodA* guarantee high expression and translation levels in *S. aureus*.

To construct the *agr* synthetic orthologous model in *B. subtilis* Δ*sigB*, the two genes *agrC* and *agrA*, which are adjacent in the operon *agrBDCA*, were cloned as a chimeric version *agrCA*. The gene *agrC* encodes the histidine kinase and the gene *agrA* encodes the cognate regulator (*Recsei et al., 1986*; *Boles and Horswill, 2008*; *Peng et al., 1988*). The resultant construct was integrated into the *amyE* neutral chromosomal locus of *B. subtilis* Δ*sigB* (strain 168). Moreover, the $P_{ica}$-*yfp*, $P_{spa}$-*yfp*, $P_{psm\alpha}$-*yfp*, $P_{psm\beta}$-*yfp*, $P_{RNAII}$-*yfp* and $P_{RNAIII}$-*yfp* transcriptional fusions were cloned into plasmid pDR183 and integrated into the neutral locus *lacA*. For transformation via double heterologous recombination, all plasmids were linearized and added to competence-induced liquid cultures of *B. subtilis* Δ*sigB* strain 168 (*Hardwood and Cutting, 1990*). Resultant colonies were verified that contained the reporter using Sanger sequencing. The synthetic model that recreates the divergent P2 and P3 promoters of *S. aureus* contains a DNA fragment of the construct of RNAII and RNAIII joined to the *cfp* and *yfp* genes divergently transcribed by the P2 (RNAII) and P3 (RNAIII) promoter, respectively. This fragment was cloned into the plasmid pDR183 and integrated into the neutral locus *lacA*. The integration of the fragment occurs by double heterologous recombination. The plasmids were linearized and added to competence-induced liquid cultures of *B. subtilis* Δ*sigB* strain 168. The resultant colonies were verified to contain the construct using Sanger sequencing.

## Staphyloxanthin extraction and quantification

For staphyloxanthin extraction, we used a protocol adapted from *Pelz et al. (2005)*. After 72 hr of growth, cells were harvested, washed once and resuspended in PBS buffer. The cell densities at $OD_{600}$ nm were measured and the samples normalized. One ml of cells was centrifuged and the pellet resuspended in 200 µl of methanol and heated at 55°C for 3 min. Samples were centrifuged to eliminate debris. Then, 200 µl of the supernatant was taken and the methanol extraction repeated. A volume of 180 µl was recovered and added to 820 µl of methanol. Absorption spectra of the methanol extracts were measured using a spectrophotometer at a peak of 465 nm, normalized and reported as relative absorbance to express the total amount of staphyloxanthin pigment.

## Atomic force microscopy indentation

Mechanical indentation via atomic force microscopy (AFM) was applied according to previous publications (*Dufrêne, 2014*; *Formosa-Dague et al., 2016*). Overnight cultures were diluted into fresh TSB or TSBMg liquid media to a final $OD_{600}$ of 0.05. Cells were grown overnight at 37°C and 220 rpm. One (1) ml of the culture was washed with sterile PBS or PBS supplemented with $MgCl_2$ final concentration 100 mM (PBSMg), depending on culture conditions and were normalized to a final $OD_{600}$ of 0.5 in PBS or PBSMg. Cells were fixed with 4% p-formaldehyde for 6 min and washed twice with 500 µl of PBS. One series of mild sonication was applied to produce a homogenous sample of

single cells. Finally, 40 µl of a dilution of 1:5 in deionized water was immobilized on poly-lysine coated microscopy slides. Samples were washed twice with Milli-Q water and allowed to dry. Samples were processed immediately after immobilization, in air and at room temperature, using a CFM conical probe AFM (Nanotec, Spain) with nominal spring constant 3 N/m and resonant frequency 75 kHz. Optical lever calibration and sensitivity was obtained by tapping the probe cantilever onto the glass surface of the slide and measuring the force response to z-piezo extension (z is vertical to the glass surface). For cell indentation, the AFM probe was placed above a cell and repeatedly pressed down onto the surface (and retracted) at 50 nm/s over distances of 100 nm, several times at several positions of the cell surface. Z position and speed of the AFM probe were controlled by a piezoelectric translator. The force response of the cell membrane was measured at three different positions for three individual cells. Young's modulus was obtained by fitting the resulting force-indentation curves for forces <10 nN, resulting in indentations <~20 nm. Best fits were produced with a modified Hertz model assuming a conical punch probe geometry.

## Purification of AIP

For purification of AIP1 from *S. aureus* strain Newman, we used a protocol that is adapted from (*MDowell et al., 2001*). To obtain an enriched fraction of AIP, a 500 ml culture of each strain was grown for 24 hr in TSB and, after the removal of bacterial cells by centrifugation, the supernatant was filtered through a 0.22 µm membrane filter and mixed 1:1 volumes with binding buffer (2% $CH_3CN$ and 1% trifluoroacetic acid). The filtered supernatant was loaded into a C18 Sep-Pak cartridge (Waters) previously stabilized with binding buffer. Elution of AIP was achieved with a 60% concentration range of $CH_3CN$ and subsequently concentrated using a *SpeedVac* system. Fresh AIP fractions were used in each experiment due to the instability of the preparation.

## Extraction and quantification of cell wall peptidoglycan

Peptidoglycan from *S. aureus* was purified using a protocol adapted from (*Bera et al., 2005*; *Peterson et al., 1978*). Cells were grown in 2 L of TSB medium and incubated overnight at 37°C with vigorous shaking. Bacteria were harvested by centrifugation (5000 × g, 4°C, 10 min), washed with cold buffer 1 (20 mM ammonium acetate, pH 4.8) and resuspended in 30 ml of buffer 1. Cell suspension was transferred to a falcon tube, centrifuged and determined the weight of the pellet. Cell pellet was resuspended in 2 ml of buffer and cells were disrupted with a bead beater (Genogrinder, SPEXsamplePrep, USA). After centrifugation, the interphase between glass beads and the foam at the top of the tube was collected and treated with 40 U of DNase, 80 U of RNase and 5 mM of $MgSO_4$ and were incubated 5 hr at 37°C. Cell walls were resuspended in 2% SDS in buffer 1 and incubated 1 hr at 65°C. The material was washed twice with distilled water and resuspended in 30 ml of buffer 1. 5% trichloroacetic acid was added to remove WTA from peptidoglycan and incubated 4 hr at 60°C. PG was then washed four to six times with cold Milli-Q water, lyophilized, and weighed. Before use, PG was resuspended in PBS buffer and sonicated on ice.

Quantification of peptidoglycan was performed using a protocol adapted from (*Nocadello et al., 2016*; *Zhou et al., 1988*). PG pellets were resuspended in 5 ml of cold buffer 1 and diluted 1:50 in a final volume of 2 ml. PG was labeled with Remazol Brilliant Blue (RBB) by incubating the samples with 20 mM RBB in 0.25 M NaOH ON at 37°C with constant shaking. The labeled samples were neutralized with HCl and pelleted by centrifugation at 14000 rpm for 20 min at room temperature. We performed intense washing using distilled water to eliminate the remaining RBB. After washing, the RBB-PG complexes were diluted 1:50 and the OD 595 nm was determined. OD 595 nm values were normalized to wet weight of each PG-isolated sample.

## Stereomicroscopy and fluorescence microscopy

Digital images of the development of *S. aureus* multicellular aggregates were captured with an Axio-CAm-HR digital camera (Carl Zeiss) using AxioVision AC Release 4.3 software (Carl Zeiss) (RRID: SCR_002677). For fluorescence microscopy, cells from the multicellular communities or from the liquid cultures were washed in PBS and resuspended in 0.5 ml of 4% paraformaldehyde solution and incubated at room temperature for 6 min. After two washing steps with PBS buffer, samples were resuspended in 0.5 ml of PBS buffer and mildly sonicated to guarantee samples of dispersed single cells. Microscopy images were taken on a Leica DMI6000B microscope equipped with a Leica

CRT6000 illumination system (Leica). The microscope was equipped with a HCX PL APO oil immersion objective with 100 × 1.47 magnification and a color camera Leica DFC630FX. Linear image processing was done using Leica Application Suite Advance Fluorescence Software (RRID:SCR_013673). The YFP fluorescence signal was detected using an excitation filter 489 nm and an emission filter 508 nm (excitation filter BP 470/40 and suppression filter BP 525/20). The RFP-mars fluorescence signal was detected using an excitation filter 558 nm and an emission filter 582 nm (excitation filter BP 546/12 and suppression filter BP 605/75). Excitation times were 567 and 875 msec, respectively. Transmitted light images were taken with 21 msec of excitation time.

To quantitatively measure cell fluorescence from microscopy images, we adapted an image protocol originally published by McCloy RA *et al.,* using ImageJ64 v1.48s (NIH, USA) (RRID:SCR_003070) (*McCloy et al., 2014*). Briefly, to quantify the number of fluorescent cells and determine their fluorescence level within a microscopy field, the overlapping image of the bright field and fluorescent channels was converted to RGB and inverted it to highlight fluorescent cells. An automatically adjusted threshold generated an image in which only fluorescent cells were represented (image A). Using the same procedure without the inversion step generated an additional image in which only non-fluorescent cells were represented (Image B). The sum between fluorescent and non-fluorescent cells from Images A and B, respectively, represented the total number of cells in the field. Quantification of fluorescence at the single cell level in the microscopy field was determined using the same software using the following commands: from the *analyze* menu, we selected *set measurements*, making sure that Area, Min and Max gray values and Mean gray value were selected. Then, we selected *analyze particles* and set: *size* in pixel unit from 20 to 200 pixels, *circularity* from 0.1 to 1.00 and *show overlay outlines*, making sure that the options *display results*, *summarize* and in situ show were selected. It is recommended to run a configuration test to set the *analyze particles* parameters that correctly cover all cells in the microscopy image. Analysis results were transferred to a Microsoft Excel sheet and to calculate the Total Cell Fluorescent (TCF) as TCF = Area of selected cell X Mean fluorescence. Results were used to generate a histogram that represents the number of particles for each mean fluorescence value. At least three independent images were analyzed for each experiment and the mean values were plotted. In average, each microscopy field comprised 500–765 cells. For analysis of overlapping signals using fluorescence microscopy, we considered signals to overlap when both signals were detected in a 3:1–1:3 range. This is the range in which green and red signals merge to yellow signal in microscopy and thus define green/red fluorescence overlap.

For thin cryosectioning of *S. aureus* multicellular communities, bright field and fluorescence images were acquired using a TCS SP5 II confocal microscope (Leica). The hardware settings included: Argon laser power at 25% and 496 nm laser intensity at 15%. Bright field images were collected using the PMT-1 Trans scan channel at 512 V with a gain offset of −0.15%. Fluorescent images were collected using the HyD-2 channel with a gain of 10 and an emission bandwidth of 500 nm for excitation and 550 nm for emission (excitation filter BP 470/40 and suppression filter BP 525/20). The acquisition mode included a xyz scan mode, with z-stacks in the z-wide mode from 4 to 8 μm. To determine the structural features of the thin sections and localize the fluorescence, a series of horizontal optical sections were collected using a z-step size of 0.3 μm. Width and height format in X and Y was set to 1024 × 1024 pixels at a scan speed of 200 Hz. Air 1 pinhole was set to automatic detection. A HCX PL APO CS 40.0 × 1.30 OIL UV objective was used for image acquisition. Digital images were captured using the Leica AF 6000 system software that is provided with the confocal microscope. All parameters were kept identical for the unlabeled control and the different labeled samples. To quantitatively measure fluorescence area from each thin cryosection image, we used ImageJ64 v1.48s and we adapted an image protocol as in (*McCloy et al., 2014*; *Gavet and Pines, 2010*; *Potapova et al., 2011*). Using this software, we quantified the bacterial aggregate area from each image of infected tissue. We quantify the proportion of fluorescent area from the total area occupied by *S. aureus* cells and referred it in percentage as relative fluorescence signal. We quantified fluorescence of three different thin cryosection samples obtained from three independent multicellular aggregates.

The infected mice organs were aseptically extracted and immersed in a solution 1:1 of PBS and paraformaldehyde 4% and left at 4°C overnight. 4 μm-thick sections were obtained using a CM 3050 s cryostat set to −20°C (Leica). These histological sections were placed on SuperFrost plus poly lysine-coated slides (Thermo) and immediately rinsed twice with PBS buffer precooled at 4°C. Then,

fixed-samples where stained with Giemsa staining solution (Sigma) including a dehydration step before the staining and a rehydration step after staining using Xylol and ethanol at 96%, 70% and 50% (*Thammavongsa et al., 2009*). Slides were immediately mounted with coverslips and processed by confocal microscopy. Histological digital images were obtained using the Diskus software (Hilgers). For fluorescence imaging, a Leica TCS SP5 II confocal microscope equipped with A HCX PL APO CS 100 × 1.47 OIL objective was used. The hardware settings included: Argon laser power at 25% and 496 nm laser intensity at 10%. Bright field images were collected using the PMT-1 Trans scan channel at 512 V. Fluorescent images were collected using the HyD-2 channel with a gain of 5 and an emission bandwidth of 500 nm for excitation and 550 nm for emission (excitation filter BP 470/40 and suppression filter BP 525/20). The acquisition mode included a xyz scan mode, with z-stacks in the z-wide mode from 4 to 8 μm. To localize fluorescence, a series of horizontal optical sections were collected using a z-step size of 0.2 μm and with an optimized system. Width and height format in X and Y was set to 1024 × 1024 pixels at a scan speed of 200 Hz. Air one pinhole was set to automatic detection. Digital images were captured using the Leica AF 6000 system software provided with the confocal microscope. All parameters remained constant during the examination of the different labeled samples. To measure fluorescence signal in infected organs, we used ImageJ64 v1.48s and we adapted an image protocol from (*McCloy et al., 2014*; *Gavet and Pines, 2010*; *Potapova et al., 2011*). Using this software, we quantified the bacterial aggregate area that exists in each one of the infected tissue images. From the area that is occupied by *S. aureus* cells, we used the same software to quantify the proportion of fluorescent area and referred in percentage relative to the total bacterial aggregate area. We quantified fluorescence of three different histological sections obtained from independent organs from three different infected mice.

## Flow cytometry

For flow cytometry analysis, cells from the multicellular communities were fixed with a treatment of 4% paraformaldehyde as mentioned above, washed and resuspended in PBS buffer. After fixation, a sonication treatment was required to separate single cells in the sample. In this case, samples were subjected to series of 25 pulses (power output 70% and cycle 0.7 s) and kept on ice. Dilution of samples 1:500 was necessary prior flow cytometry analyses. For YFP fluorescence, a laser excitation of 488 nm coupled with 530/30 and 505LP sequential filter was used. The photomultiplier voltage was set at 777 V.

## Fluorescence-activated cell sorting

To obtain samples enriched in BRcells or DRcells, we used single-labeled *Staphylococcus aureus* multicellular communities between day 4 and 5 of development. These strains were labeled to differentiate cells expressing the extracellular matrix-production reporter (P$_{ica}$-*yfp*) or the detachment/virulence reporter (P$_{psmα}$-*yfp*). Multicellular communities were scraped from the TSBMg plates and immediately resuspended in *RNAlater* (Qiagen) in 1.5 ml RNAse-free Eppendorf tubes, in order to fix the cell fluorescence and at the same time preserve the RNA within the cells. Previous reports (*Rosenberg et al., 2003*) and fluorescence microscopy experiments performed in our laboratory (data not shown) showed that the fixing procedure of these multicellular communities in *RNAlater* had no effect in the conservation of the fluorescence when compared to cells fixed using 4% paraformaldehyde. Multicellular communities were disrupted in the *RNAlater* by extensive pipetting, followed by one series of mild sonication as mentioned above, and previously treating the sonicator with RNaseZap RNase Decontamination Solution (Life Technologies) (RRID:SCR_008817). All procedures were performed on ice. After sonication, samples were immediately processed using FACS. Cell fixation and subsequent mild sonication allowed cell separation without affecting cell integrity. For the sorting procedure, 50 μl of the cell suspension was resuspended in 10 ml of filtered and autoclaved PBS buffer prepared in DEPC-treated water. This cell suspension was sonicated, changing cycles from 70% to continuous (100%) and performing 1 round of 20 s. Immediately, cells were FACS-sorted based on their fluorescence intensity in a FACS Aria III (Becton Dickinson) (RRID:SCR_008418) using the following parameters: Nozzle size of 70 microns, FITC/Alexa Fluor 488 nm laser, a 530/30 nm filter for data collection and a 502 LP mirror. Flow cytometry parameters were set as: SSC 341 V with a threshold of 500, FSC 308 V with a threshold of 500, FITC 769 V and a variable Flow Rate to guarantee that the number of events per second never exceeded 1500; hence, the

Sorting Efficiency never dropped from 97%. These data were analyzed using the BDIS FACS Diva software version 7.0 provided with the FACS Aria III. Sorting was performed in the Precision Mode set to 'Single Cell' in a first round, followed by a second sorting round set to 'Purity'. Using the sorting Precision Mode, we recovered approx. 25 million cells of each subpopulation (fluorescent cells) and their respective non-fluorescent counterparts, based on manually established Target Gates P1 for highly fluorescent cells and P2 for non-fluorescence cells. Once sorted (approximately 5 million cells per 15 ml tube), cells were immediately quick-frozen by immersing the tubes in liquid nitrogen prior to ultra-freezing until sorting was completed.

## RNA isolation

For the FACS-sorted bacterial cells, the ultra-frozen samples were thawed using a 37°C water bath. The volume was poured in a vacuum filter system provided with a 47 mm filter diameter and 0.45 µm pore-size. The equipment was previously sterilized using 75% ethanol in DEPC-treated water, followed by two DEPC-treated water rinses and finally UV light for 180 s and then precooled at −20°C. Filters containing each of the sorted samples were individually ground using liquid Nitrogen in an RNAse-free, sterile precooled mortar. The powder was carefully scraped out from the mortar and placed in a 2 ml RNAse-free Eppendorf tube and the RNA isolated as described below.

To isolate RNA from *S. aureus* infected organs, 450 µl of the homogenized organs were incubated for 15 min on ice with 50 µl of *RNAlater* and 5 µl of Triton X100 briefly vortexing every 5 min to lyse the murine cells. After lysis, RNA was isolated as described below. To isolate RNA from *S. aureus* infected organs, 450 µl of the homogenized organs were incubated for 15 min on ice with 50 µl of RNAlater and 5 µl of Triton X100 briefly vortexing every 5 min to lyse the murine cells. After lysis, RNA was isolated as described below.

One volume of TE lysis buffer (Tris 20 mM pH 7.5, EDTA 10 mM) prepared in DEPC-prepared water and 25 µl of Lysostaphin (1 mg/ml) were added and taken to a hybridization oven (with rotation), pre-warmed at 37°C. Samples were incubated for 30 min at 37°C prior to transferring the whole material to a new tube for mechanical lysis. This was performed using FastPrep Lysing Matrix glass beads in a Fast Prep Shaker (MP Biomedicals) (RRID:SCR_013308) set at 6500 rpm for 50 s. Samples were lysed using 2 cycles of 50 s. The lysate was transferred to a new 2 ml RNAse-free Eppendorf tube and centrifuged for 10 min at 14000 rpm and 4°C to remove the filter and beads. The supernatant (700 µl) was recovered and used for RNA isolation using the standard hot phenol methodology with some modifications. The sample recovered from was mixed with 60 µl of 10% SDS and incubated at 64°C for 2 min. 66 µl of sodium acetate 3 M pH 5.2 and 750 µl of phenol Roti-Aqua (Carl Roth) were added and the mixture was incubated at 64°C for 6 min with mixing every 30 s. The sample was centrifuged for 10 min at 13000 rpm and 4°C and the upper aqueous layer was transferred to a 2 ml Phase Lock Gel Heavy (PLGH) tube (5Prime). Then, 750 µl of Chloroform was added and the sample was centrifuged for 12 min at 13000 and 15°C. The aqueous layer was transferred to a new tube and mixed with 2 volumes of a 30:1 ethanol and sodium acetate 3 M pH 6.5 mix. Additionally, 0.5 µl of GlycoBlue co-precipitant (15 mg/ml) (Life Technologies) was added. This mix was left at −20°C overnight. The following day, the sample was removed from storage at −20°C, centrifuged for 30 min at 13000 rpm and 4°C and the pellet was washed with 300 µl of precooled 75% ethanol. After washing, the total RNA was resuspended in 42 µl of RNAse-free water (Qiagen) were added and the sample was incubated at 65°C at 1000 rpm for 5 min prior storage in ice. To remove any DNA traces, the isolated RNA was treated (one to three times, depending on the sample) with 4 Units of RNase-free DNase I (Thermo), 10 Units of SUPERase In RNase Inhibitor (Life Technologies) and incubated for 45 min at 37°C. To remove the DNase I, 50 µl of RNAse-free water and 100 µl of Roti-Aqua-P/C/I (Phenol, Chloroform, Isoamyl alcohol 25:24:1 pH 4.5–5) (Carl Roth) were added to the reaction tube, mixed, transferred to a 2 mL PLGH tube and centrifuged for 12 min at 13000 rpm and 15°C. The sample was mixed with 2.5 volumes of the 30:1 ethanol and sodium acetate with 0.5 µl of GlycoBlueTM, which led to RNA precipitation when stored at −20°C overnight. On the following day, samples were centrifuged for 30 min at 13000 rpm and 4°C, washed with 200 µl of 75% ethanol and the pellets were dried and resuspended in 50 µl of DEPC-water. To remove phenol residues, 50 µl of RNAse-free water and 100 µl of Chloroform (Carl Roth) were added to the reaction tube, mixed by inversion for 1 min, transferred to a 2 mL PLGH tube and centrifuged for 12 min at 13000 rpm and 15°C. The sample was mixed with 2.5 volumes of the 30:1

ethanol and sodium acetate with 0.5 µl of GlycoBlueTM, which led to RNA precipitation when stored at −20°C overnight. On the following day, samples were centrifuged for 1 hr at 13000 rpm and 4°C, washed with 200 µl of 75% ethanol and the pellets were dried. This RNA was resuspended in 22 to 44 µl of RNAse-free water and then incubated at 65°C at 1000 rpm for 5 min. To assess the concentration and purity of the total RNA, $OD_{260}$ was measured using a Nanodrop (Thermo) and the $OD_{260}/OD_{280}$ ratio and the $OD_{260}/OD_{230}$ ratio determined.

## RNA-Seq library construction, sequencing and quantitative-PCR analysis

The cDNA prepared was strictly strand-specific, allowing transcriptome sequencing and expression profiling in both the forward and reverse strands. The combined-length of the flanking sequences was 100 bases. The cDNA is generated and size fractionated by preparative gel electrophoresis or by using the LabChip XT fractionation system from Caliper/PerkinElmer in order to obtain cDNA fractions, optimally suited for the different NGS systems. For this, the RNA samples were poly (A)-tailed using a poly(A) polymerase. The 5'-PPP were removed using tobacco acid pyrophosphatase (TAP) followed by the ligation of the RNA adapter to the 5'-monophosphate of the RNA. First-strand cDNA synthesis was performed with an oligo (dT)-adapter primer and the M-MLV reverse transcriptase. The resulting cDNA was PCR-amplified to reach a concentration of 20–30 ng/µl using a high fidelity DNA polymerase. The cDNA was purified using the Agencourt AMPure XP kit (Beckman) (RRID:SCR_008940) and was analyzed by capillary electrophoresis. The primers used for PCR amplification were designed for TruSeq sequencing according to the instructions of Illumina. The following adapter sequences flank the cDNA inserts: TruSeq_Sense: 5'-AATGATACGGCGACCACCGAGATC TACACTCTTTCCCTACACGACGCTCTTCCGAT-T-3' TruSeq-Antisense NNNNNN (NNNNNN = B arcode) 5'-CAAGCAGAAGACGGCAT ACGAGATNNNNNNGTGACTGG-AGTTCAGACGTGTGCTC TTCC-GATC(dT25)−3'.

For quantification of gene expression, total RNA was reverse-transcribed using hexameric random primers followed by quantitative real-time PCR (qRT–PCR) using the *SsoAdvanced SYBR Green Supermix* (Bio-Rad) (RRID:SCR_013553), following manufacturer's instructions. Primer pairs used are described in the *Supplementary file 2*. Gene expression was normalized to *gyrA*/*gapA* expression and expression fold changes were calculated using the $2^{−\Delta\Delta Ct}$ method. These qRT-PCR experiments were performed following the standard MIQE guidelines for publication of qRT-PCR experiments (*Bustin et al., 2009*).

## Bioinformatics analysis

The pooled sequence reads were de-multiplexed and the adapter sequences were removed. After that, the reads in Fastq format were quality trimmed using fastq_quality_trimmer (from the FastX suite version 0.0.13) (RRID:SCR_005534) with a cut-off Phred score of 20 and converted to Fasta format using Fastq_to_Fasta (also from the FastX suite). The reads were processed, which included poly(A) removal, size filtering (minimum read length of 12 nucleotides after clipping), statistics generation, coverage calculation and normalization, which were performed with the RNA-analysis pipeline READemption version 0.3.3. READemption uses segemehl version 0.1.7 for the read alignment to the reference and DESeq 1.18.0 (*Anders and Huber, 2010*) (RRID:SCR_000154) for the differential gene expression analysis. The reference genome NC_009641 was taken from the NCBI database for the purpose of alignment and gene-quantification. DESeq calculates statistically significant expression fold change and their log2 values by computing the ratio of normalized read counts of each gene in two libraries. The genes with log2fold value higher than 1.5 and lower than −1.5 were selected as up- and down-regulated gene-sets, respectively. Scatter plots for the visualization of sample correlation were generated using matplotlib 1.4.2. 2. Datasets used for gene ontology for functional classification of genes differentially expressed, hierarchical clustering and the calculation of the hypergeometric probability for genes differentially expressed are presented in *Figure 6— source data 2* to 4.

## Genome-wide analyses to compare BRcells and DRcells

We estimated the difference between the reference genome and the sequences from the libraries (total number of reads in the libraries range from ~6–11 million with an average of 100 nucleotides per read). Libraries were analyzed using SAMtools (mpileup command) and variant calling by

BCFtools (RRID:SCR_005227). The variants were quantified for the reads mapped to the coding regions of the reference genome using the standard quality score >40 (base call accuracy 99.99%). SAM (Sequence Alignment/Map) tools enables quality checking of reads, and automatic identification of genomic variants (*Li et al., 2009b*). A high quality score means higher number of sequences showing a particular kind of variation. Quality score is proportional to the number of reads mapped to a gene that account for the variation. The BCFtools were used for the variance calling to identify general variances in each library and compared with each other to determine the existence of library-specific variations (*Li, 2011*). Using this approach, we detected a minimum of 99.999934% genome similarities at the nucleotide level between distinct cell types.

## Statistical analysis

All statistical analyses were performed using the software Prism 6 (version 6.0 f, GraphPad) (RRID: SCR_002798). Graphs represent data from at least three independent experiments with at least three independent technical replicates for experiment. Error bars represent standard deviation (mean ±SD). For the analysis of experiments with two groups, the parametric unpaired two-tailed Student's t-test with Welch's correction was done and, the non-parametric unpaired Mann-Whitney test were done. For the analysis of experiments with three or more groups, the parametric one-way ANOVA test was done. *Post hoc* analysis included multiple comparisons Tukey's test, Dunnett's test or Dunn's tests, depending on the data set. Differences were considered significant when $p$ value was smaller than 0.05. Statistical significance: ns = not statistically significant, *$p<0.05$, **$p<0.01$, ***$p<0.001$.

## Mouse infection studies

All animal studies were approved by the local government of Lower Franconia, Germany (license number 55.2-DMS-2532-2-57 and were performed in strict accordance with the guidelines for animal care and animal experimentation of the German animal protection law and directive 2010/63/EU of the European parliament on the protection of animals used for scientific purposes. Female BALB/c mice (16 to 19 g) were purchased from Charles River (Charles River Laboratories, Erkrath, Germany) (RRID:SCR_003792), housed in polypropylene cages and supplied with food and water *ad libitum*. The different *S. aureus* strains used were cultured for 18 hr at 37°C on BHI medium. Subsequently, cells were collected and washed three times with PBS and diluted to reach an $OD_{600\ nm} = 0.05$. Viable cell counts were determined by plating dilutions of the inoculum on TSB agar plates. To compare the representation of each cell type of *S. aureus* in the bacterial communities that colonize the distinct organs, three cohorts of 3 mice for the unlabeled strain and six mice for each labeled strains ($P_{ica}$-*yfp* and $P_{psm\alpha}$-*yfp*) were infected with 150 µl of cultures of *S. aureus*, containing $1 \times 10^7$ cells via tail vein injection. Each strain was used to infect one cohort of mice. Infections were allowed to progress until severe infection signs occurred or to the endpoints of 24 hr, 48 hr, 72 hr and 96 hr after mice challenging. Animals were sacrificed when they met the following criteria: 1) loss of at least 20% of body weight; 2) loss of at least 15% of body weight and ruffled fur; 3) loss of at least 10% of body weight and hunched posture; or 4) 24 hr, 48 hr, 72 hr and 96 hr time points after infection. Infected kidneys and hearts were aseptically harvested and processed for thin sectioning, histology and confocal microscopy, as described above. Organs were also used to calculate bacterial burden calculated as CFU/g of organ, as proposed by *Marincola et al. (2012)*. For this purpose, kidneys and hearts were homogenized in 2 ml of sterile PBS using GentleMACS™ M Tubes (Miltenyi Biotec) (RRID:SCR_008984) and serial dilutions from $10^{-2}$ to $10^{-8}$ of the organ homogenates were immediately plated on TSB plates and incubated at 37°C for 24 hr, To compare the representation of the *S. aureus* low-*tagB* and high-*tagB* strains colonizing the kidneys, heart and liver, four cohorts of 5 mice each, were infected with 150 µl of cultures of *S. aureus* and organs were processed as described above. Three days after bacterial challenge all mice were euthanized, organs were aseptically harvested and processed as described above, CFU determined and RNA isolated.

## Mathematical modeling of *agr* bimodal behavior

To study the dynamic processes of the synthetic network, we mathematically described the reactions of the genetic circuitry of the orthologous system. In this circuit, the phosphorylation state is only reached inside the cell after the concentration of AIP is above the threshold. The complex AgrA~P is

the transcription factor that upregulates the expression of P2 promoter (also called $P_{RNAII}$) responsible for activation of the positive feedback loop. Moreover, AgrA~P upregulates the expression of the rest of the promoters that were used in this work; P3 (Also called $P_{RNAIII}$), $P_{psm\alpha}$, $P_{psm\beta}$. $P_{ica}$ and $P_{spa}$ promoters are not directly regulated by AgrA~P and were therefore used as negative controls. The reactions that define the former genetic network are the following:

$$\text{Activation/deactivation}: \quad AgrA^P + P_{agrA} \underset{k_{-1}}{\overset{k_1}{\rightleftharpoons}} P^a_{agrA} \tag{1}$$

$$\text{Fast transcription}: \quad P^a_{agrA} \xrightarrow{k_2} P^a_{agrA} + mAgrA \tag{2}$$

$$\text{Slow transcription}: \quad P_{agrA} \xrightarrow{k_3} P_{agrA} + mAgrA \tag{3}$$

$$\text{Translation}: \quad mAgrA \xrightarrow{k_4} mAgrA + AgrA \tag{4}$$

$$\text{Complex formation}: \quad AgrA + P \xrightarrow{k_5} AgrA^P \tag{5}$$

$$\text{Creation(appearance)}: \quad \phi \xrightarrow{k_6} P \tag{6}$$

$$\text{Activation/deactivation}: \quad AgrA^P + P_x \underset{k_{-7}}{\overset{k_7}{\rightleftharpoons}} P^a_x \tag{7}$$

$$\text{Fast transcription}: \quad P^a_x \xrightarrow{k_8} P^a_x + mYFP \tag{8}$$

$$\text{Slow transcription}: \quad P_x \xrightarrow{k_9} P_x + mYFP \tag{9}$$

$$\text{Translation}: \quad mYFP \xrightarrow{k_{10}} mYFP + YFP \tag{10}$$

$$\text{Degradation (disappearance)}: \quad mAgrA \xrightarrow{k_{11}} \phi \tag{11}$$

$$\text{Degradation (disappearance)}: \quad AgrA \xrightarrow{k_{12}} \phi \tag{12}$$

$$\text{Degradation (disappearance)}: \quad AgrA^P \xrightarrow{k_{13}} \phi \tag{13}$$

$$\text{Degradation (disappearance)}: \quad P \xrightarrow{k_{14}} \phi \tag{14}$$

$$\text{Degradation (disappearance)}: \quad mYFP \xrightarrow{k_{15}} \phi \tag{15}$$

$$\text{Degradation (disappearance)}: \quad YFP \xrightarrow{k_{16}} \phi \tag{16}$$

where $k_1$ and $k_{-1}$ are the binding and unbinding rates of AgrA~P to P2, $k_2$ is the transcription rate of P2 once AgrA~P binds the promoter, $k_3$ is the basal transcription rate of P2, $k_2$ and $k_3$ produce the mRNA of *agrA*, $k_4$ is the translation rate of AgrA protein, $k_5$ is the phosphorylation rate of AgrA, $k_6$ is the availability rate of phosphate in the system, $k_7$ and $k_{-7}$ are the binding and unbinding rates of

AgrA~P to the different promoters ($P_x$), $k_8$ is the transcription rate of $P_x$ once AgrA~P binds to the promoter ($P^a_x$), $k_9$ is the basal transcription rate of $P_x$, $k_{10}$ is the translation rate of the YFP protein and $k_{11}$ to $k_{16}$ are the degradation rates of mRNAs and proteins involved in this system.

Deterministic modeling using differential equations pointed to a quasi-steady state assumption (*Murray, 2002*), which we used to identify the elements responsible for the behavior of the system. The resulting equations are:

$$\frac{dAgrA^P}{dt} = \frac{k_6 \cdot AgrA}{AgrA + \delta} - k_{13} \cdot AgrA^P \tag{17}$$

$$\frac{dAgrA}{dt} = \frac{\alpha_1 + \beta_1 \cdot AgrA^P}{\gamma_1 + AgrA^P} - \frac{k_6 \cdot AgrA}{AgrA + \delta} - k_{12} \cdot AgrA \tag{18}$$

$$\frac{dYFP}{dt} = \frac{\alpha_2 + \beta_2 \cdot AgrA^P}{\gamma_2 + AgrA^P} - k_{16} \cdot YFP \tag{19}$$

where $\alpha_1 = k_3k_{-1}k_4Pt_{agrA}/k_1k_{11}$, $\beta_1 = k_2k_4Pt_{agrA}/k_{11}$, $\gamma_1 = k_{-1}/k_1$, $\delta = k_{14}/k_5$, $\alpha_2 = k_9k_{-7}k_{10}Pt_x/k_7k_{15}$, $\beta_2 = k_8k_{10}Pt_x/k_{15}$, $\gamma_2 = k_{-7}/k_7$ and $Pt_i = P_i + P^a_i$ with i = [*agrA*, *x*].

The following values are used to run the Gillespie algorithm of the full model (reactions 1–16). We focused our first set of simulations on the dynamics that affect YFP directly, to further study the behavior of the system when considering variations in AgrA and AgrA~P. The first set of simulations allowed us to find parameters that show a fixed value among all of the experiments independently on the promoter that is under consideration. These parameters are shown here:

| Parameter | Meaning | Value |
|---|---|---|
| $k_1$ | Binding rate | 0.01 molecules/h |
| $k_{-1}$ | Unbinding rate | 2/h |
| $k_2$ | Transcription rates | 500/h |
| $k_3$ | Basal transcription rate | 50/h |
| $k_9$ | Basal transcription rate | 90/h |
| $k_4$, $k_{10}$ | Translation rates | 50/min |
| $k_5$ | Phosphorylation rate | 0.05/molecules/hour |
| $k_6$ | Entry rate | 40/molecules/hour |
| $k_{11}$, $k_{15}$ | Degradation rates | 10/hour |
| $k_{12}$ | Degradation rate | 0.05/hour |
| $k_{13}$ | Degradation rate | 0.1/hour |
| $k_{14}$, $k_{16}$ | Degradation rates | 1/hour |

Consequently, the values of these parameters did not change through the rest of the simulations that aimed to characterize the kinetics of each particular promoter. Initial simulations assumed the high stability of AgrA~P to test the response of the system to saturated levels of AgrA~P. The following simulations assumed unstable AgrA~P. The specific rates to simulate each promoter were defined as follows:

| Parameter | Meaning (rate) | $P_{RNAIII}$ | $P_{psm\alpha}$ | $P_{RNAII}$ | $P_{psm\beta}$ | $P_{RNAIII-dual}$ | $P_{RNAII-dual}$ |
|---|---|---|---|---|---|---|---|
| $k_8$ | Transcription | 300 | 300 | 450 | 470 | 300 | 450/hour |
| $k_7$ | Binding | 3 | 6.5 | 10 | 20 | 12 | $40 \times 10^{-4}$* |
| $k_{-7}$ | Unbinding | 0.08 | 0.08 | 0.1 | 0.1 | 0.08 | 0.1/hour |

*molecules$^{-1}$ hour$^{-1}$

The simulations that involved an unstable AgrA~P established values of $k_5 = 0.2 \times 10^{-4}$ and $k_6 = 5$. Using these values, we resolved a 3-mode decision-making model. Further simulations that

involved changeable values of $k_6$ (phosphate availability) revealed that the activation rate of P2 and P3 occurs within a range of 0 and a maximum value of saturation of 40. The values for the rest of the parameters were selected from standard numbers obtained from previously reported studies (*Andersen et al., 1998*; *Balagaddé et al., 2008*; *Ben-Tabou de-Leon and Davidson, 2009*; *Dublanche et al., 2006*; *Goñi-Moreno and Amos, 2012*).

## Mathematical modeling of multicellular communities development

We used a so-called reaction diffusion system, which not only describes the changes of concentrations and density in time to any type of reaction but also their spread in space. In our case, we used a model with two spatial dimensions. Our system consisted of four equations, describing how the concentrations of nutrients, AIP and the density of replicative and non-replicative bacteria evolved in time. In the following, $n(x, t)$ denotes the nutrient concentration, $b(x, t)$ the density of replicative bacteria, $s(x, t)$ the density of non-replicative bacteria and $q(x, t)$ the concentration of AIP. We assumed that nutrients and AIP underlie diffusion. Diffusion parameters are denoted by $d_n$ and $d_q$, respectively. In the case of the replicative bacteria, the diffusion coefficient depends on nutrient concentration and the density of replicative bacteria. It is of the form $d_b = \sigma nb$ where $\sigma$ also contains a stochastic fluctuation of random movement. This form for the diffusion of active bacteria cells was chosen according to the model developed by *Kawasaki et al. (1997)*. We assumed that non-replicative bacteria are not able to diffuse on their own and their diffusion is driven by movement of the replicative cells. The diffusion of non-replicative bacteria depends on the density of replicative bacteria. If there are more replicative bacteria, the non-replicative cells will be pushed in a type of diffusion. This effect is limited. The diffusion coefficient for the non-replicative bacteria is assumed to be of the form:

$$d_s = \tau \frac{b}{b_s + b}$$

where $b_s$ and $\tau$ are constant. The replicative bacteria proliferate by consuming nutrients. In our system of equations, the consumption rate of the nutrients was given by $G_1 f(n, b)$ and the bacterial growth was described by the term $G_2 f(n, b)$ where $G_2/G_1$ was the conversion rate of consumed nutrients to bacterial growth. We assumed $f(n, b)$ to be of the form:

$$f(n, b) = \frac{bn}{1 + \gamma n}\left(1 + \frac{1}{q_m + \delta q}\right)$$

Here, we chose a Monod growth term (*Monod, 1949*) to describe the increase in the concentration of replicative bacteria in relation to nutrient consumption. It reproduces the fact that a high nutrient concentration will cause a faster increase in the concentration of replicative bacteria but that these bacteria cannot reproduce infinitely fast. An additional factor accounts for the effect of the quorum sensing signal. If the concentration of active bacteria is already high, a high concentration of AIP slows down the conversion process. We furthermore assumed that there is only a transition from replicative to non-replicative cells. This process is described by a term of the form:

$$\varepsilon a(b, n) = \varepsilon \frac{b}{\left(1 + \frac{b}{a_1}\right)\left(1 + \frac{n}{a_2}\right)}$$

This choice is in agreement with (*Matsushita et al., 1999*). The equation number four described the concentration of AIP. We considered that the diffusion and the increase in the concentration of AIP are related with the concentration of replicative bacteria but we also considered that there is degradation process of AIP. The degradation of AIP is described by $d_q$, whereas the production of AIP typically has two levels in a Hill type function (*Gustafsson et al., 2004*) with Hill coefficient 2 to reflect bistability. This is caused by a positive feedback including nonlinearity in the underlying regulation system. The low production rate of AIP is denoted by $p_1$ while the increased production rate is denoted by $p_2$. The threshold between the low and the increased production is denoted by $q_{thr}$. For more details see (*Müller et al., 2006*). With the above-mentioned terms we achieved the following system of equations, which are able to represent the growth dynamics of *S. aureus* multicellular aggregates:

$$\frac{\partial n}{\partial t} = d_n \nabla^2 d - G_1 \frac{bn}{1+\gamma n} \left(1 + \frac{1}{q_m + \delta q}\right) \tag{20}$$

$$\frac{\partial b}{\partial t} = \nabla(\sigma \mathrm{nb} \nabla \mathrm{b}) + G_2 \frac{bn}{1+\gamma n} \left(1 + \frac{1}{q_m + \delta q}\right) - \varepsilon \frac{b}{\left(1 + \frac{b}{a_1}\right)\left(1 + \frac{n}{a_2}\right)} \tag{21}$$

$$\frac{\partial s}{\partial t} = \nabla\left(\tau \frac{b}{b_s + b} \nabla s\right) + \varepsilon \frac{b}{\left(1 + \frac{b}{a_1}\right)\left(1 + \frac{n}{a_2}\right)} \tag{22}$$

$$\frac{\partial q}{\partial t} = d_q \nabla^2 q + \left(p_1 + p_2 \frac{q^2}{q_{thr}^2 + q^2}\right) b - \mu_q q \tag{23}$$

It remains necessary to define the initial and boundary conditions. Since the bacteria grow on an agar plate, we chose no-flux boundary conditions, e.g. $\frac{\partial n}{\partial x}|_{\partial\Omega} = 0$, where $\Omega$ denotes the area of the agar plate. The agar has the same concentration of nutrients throughout and therefore we defined the initial condition for the nutrients as constant over the entire domain, e.g. $n(x,0) = n_0$ for $x \in \Omega$. The replicative bacteria are set on the agar as a drop in the center. At this stage, neither non-replicative bacteria nor AIP exist and thus the initial conditions read $b(x,0) = b_M \exp\left(-\frac{x^2+y^2}{6.25}\right)$ with a compact support, $s(x,0) = 0$ and $q(x,0) = 0$. The (non-dimensionalized) parameters chosen for this simulation are given by:

$$\sigma = 0.5; G_1 = G_2 = G = 7; \varepsilon = 5; \tau = 0.25; q_m = 0.3; d_n = d_q = 1; z = 1; \rho = 1;$$

$$\gamma = 1; b_s = 2; \delta = 1; q_{thr} = 1; \mu = 1; n_0 = 1.11; p_1 = p_2 = 1;$$

$$a_1 = 2400; a_2 = 120.$$

Our model was able to capture differences in texture on the surfaces of different *S. aureus* mutants. For the simulation of mutants, we only changed the parameter value, which corresponds to the modified gene and phenotype for the mutant, the other parameter values were maintained from the wild type, as follows:

| Background | $\sigma$ | G | $p_1$ | $p_2$ |
|---|---|---|---|---|
| Wild type | 0.5 | 7 | 1 | 1 |
| spa | 0.5 | 3 | 1 | 1 |
| ica | 0.75 | 2 | 1 | 1 |
| psmα | 1 | 8 | 1 | 1 |
| psmβ | 1 | 4 | 1 | 1 |
| agr | 1 | 5 | 0 | 0 |
| spa ica | 1 | 2 | 1 | 1 |

The non-mentioned parameters were maintained at the same value as the wild type strain. In the case of the $\Delta spa$ mutant, as influencing the biofilm, we modified the biofilm production rate constant $G$ from 7 to 3. The mutants $\Delta psm\alpha$ and $\Delta psm\beta$ differ from the wild type also by their ability to move on; therefore apart from $G$ also the parameter $\sigma$ was modified. To obtain the *agr* mutant not only a simple change in the parameters is needed, but also a combination of different mutant behaviors since the *agr* system influences several components of biofilm production and properties. Especially the ability of the model to produce AIP is lost, expressed by setting $p_1$ and $p_2$ to 0.

## Acknowledgements

We thank M Luna (CSIC, Spain) for assistance with atomic force microscopy and C Mark (CSIC, Spain) for editorial assistance. JCGB was supported by GSLS from the University of Würzburg. Sequencing files have been deposited in the NCBI GEO Series (accession number GSE69835).

## Additional information

### Funding

| Funder | Grant reference number | Author |
| --- | --- | --- |
| H2020 European Research Council | 335568 | Daniel Lopez |
| Deutsche Forschungsge-meinschaft | Lo1804-2/2 | Daniel Lopez |
| Ministerio de Economía y Competitividad | BFU2014-55601-P | Daniel Lopez |

The funders had no role in study design, data collection and interpretation, or the decision to submit the work for publication.

### Author contributions

Juan-Carlos García-Betancur, Angel Goñi-Moreno, Thomas Horger, Melanie Schott, Julian Eikmeier, Investigation, Methodology; Malvika Sharan, Software, Formal analysis; Barbara Wohlmuth, Software, Formal analysis, Supervision; Alma Zernecke, Conceptualization, Formal analysis; Knut Ohlsen, Conceptualization, Data curation; Christina Kuttler, Conceptualization, Software, Formal analysis; Daniel Lopez, Conceptualization, Supervision, Funding acquisition, Investigation, Writing—original draft, Project administration, Writing—review and editing

### Author ORCIDs

Juan-Carlos García-Betancur http://orcid.org/0000-0003-3371-3384
Julian Eikmeier http://orcid.org/0000-0002-1576-1174
Daniel Lopez http://orcid.org/0000-0002-8627-3813

### Ethics

Animal experimentation: All animal studies were approved by the local government of Lower Franconia, Germany (license number 55.2-DMS-2532-2-57 and were performed in strict accordance with the guidelines for animal care and animal experimentation of the German animal protection law and directive 2010/63/EU of the European parliament on the protection of animals used for scientific purposes.

### Decision letter and Author response

Decision letter https://doi.org/10.7554/eLife.28023.031
Author response https://doi.org/10.7554/eLife.28023.032

## Additional files

### Supplementary files

• Supplementary file 1. List of strains used in this study.
DOI: https://doi.org/10.7554/eLife.28023.025

• Supplementary file 2. List of plasmids and plasmids used in this study.
DOI: https://doi.org/10.7554/eLife.28023.026

• Transparent reporting form
DOI: https://doi.org/10.7554/eLife.28023.027

## Major datasets

The following dataset was generated:

| Author(s) | Year | Dataset title | Dataset URL | Database, license, and accessibility information |
|---|---|---|---|---|
| Garcia-Betancur JC, Goñi- Moreno A, Horger T, Schott M, Sharan M, Eikmeir J, Wohlmuth B, Zernecke A, Ohlsen K, Kuttler C and Lopez D | 2017 | Cell Differentiation Defines Acute and Chronic Infection Cell Types in Staphylococcus aureus | https://www.ncbi.nlm. nih.gov/geo/query/acc. cgi?acc=GSE69835 | Publicly available at the NCBI Gene Expression Omnibus (accession no. GSE69835) |

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
