## [Decision Letter]

Thank you for submitting your article "Cell Differentiation Defines Acute and Chronic Infection Cell Types in *Staphylococcus aureus*" for consideration by *eLife*. Your article has been reviewed by two peer reviewers, one of whom, Michael S Gilmore (Reviewer #1), is a member of our Board of Reviewing Editors, and the evaluation has been overseen by Gisela Storz as the Senior Editor. The following individuals involved in review of your submission have agreed to reveal their identity: Friedrich Gotz (Reviewer #3).

The reviewers have discussed the reviews with one another and the Reviewing Editor has drafted this decision to help you prepare a revised submission.

Summary:

The manuscript "Cell Differentiation Defines Acute and Chronic Infection Cell Types in *Staphylococcus aureus*" examines the differentiation of S. aureus into biofilm related or dispersal related cells. The authors find a bimodal population of S. aureus cells as determined by the measurement of florescence transcriptional reporters under the control of either biofilm related gene promoters or dispersal related gene promoters. This bimodality is controlled by agr quorum-sensing system, and can be recapitulated in another organism, *Bacillus subtilis*. This shift between BRcells and DRcells can be influenced by environmental signals, and magnesium is identified as a major signal in the shift from DRcells to BRcells, via the stabilization of the cell wall by interaction with the teichoic acids. This leads to an induction of sigmaB stress response, the downregulation of the agr quorum-sensing system and shift to the BRcell state.

Supporting the model that these different cell populations are important for causing diverse diseases types in vivo, the authors report a change in bacterial load in the organs with high or low Mg^2+^ concentrations (kidneys vs heart) in cells with altered BR or DRcell populations (via genetic manipulation of TA levels via TagB expression). They also report a higher fluorescence signal of BRcells within the reportedly high Mg^2+^ kidney tissue and higher DRcell reporter signal in the low Mg^2+^ heart tissue.

Overall, this paper advances our understanding of the way in which S. aureus is able to respond to environmental signals to establish infections in different sites within the body, with an extensive analysis of the mechanism by which this bimodality is controlled.

Essential revisions:

1) In Figure 3 the authors use cell lines which they suggest have altered expression of TagB and therefore altered the teichoic acid levels of the bacterial cell wall. It does not appear that these strains have been previously published, and therefore it is important that the authors show that these strains do in fact have altered expression of TagB (either through qRT PCR or protein level analysis) and/or altered teichoic acid levels in the cell wall. This is particularly important as these strains are used in the final animal experiments, where a great deal of significance of the paper lies.

2) The pivotal finding of the paper is that in different sites of infection within the animal model S. aureus cells are in either the BRcell or DRcell state. As such, Figure 7 is critical to supporting this claim and would be improved with additional examples and clearer labeling of immune cell infiltrates. The low magnification images shown in Figure 7) do not contribute importantly to comprehension of the figure, and do not seem to highlight immune cell infiltrates. Including additional images in this figure (or as a supplement) displaying this is recommended. Additionally, it is unclear exactly how the authors performed the "semi-quantitative" imaging of the BRcell and DRcell reporter cells. It states that it was performed on three images, were these from independent animals? The same kidney or multiple infected kidneys? Additional explanation of this quantification method and representative fluorescence images are warranted here as well.

3) The overall organization of the results and figures made reading and interpreting the data more difficult than necessary. The results are not discussed in the order of the figure panels, and the supplemental figures were out of order in the manuscript. Organizing the figure panels in the order they are discussed in the Results section (or vice versa) will greatly improve the reading experience.

4) There are quite a number of publications dealing with heterogeneity and adaptation in S. aureus subpopulations and which regulators control heterogeneity – including the following. The findings should be placed into context.

Audretsch, C., D. Lopez, M. Srivastava, C. Wolz & T. Dandekar, (2013) A semi-quantitative model of Quorum-Sensing in Staphylococcus aureus, approved by microarray meta-analyses and tested by mutation studies. Mol Biosyst 9: 2665-2680.

Geiger, T., B. Kastle, F.L. Gratani, C. Goerke & C. Wolz, (2014) Two small (p)ppGpp synthases in Staphylococcus aureus mediate tolerance against cell envelope stress conditions. J Bacteriol 196: 894-902.

Geiger, T. & C. Wolz, (2014) Intersection of the stringent response and the CodY regulon in low GC Gram-positive bacteria. Int J Med Microbiol 304: 150-155.

George, S.E., T. Nguyen, T. Geiger, C. Weidenmaier, J.C. Lee, J. Liese & C. Wolz, (2015) Phenotypic heterogeneity and temporal expression of the capsular polysaccharide in Staphylococcus aureus. Mol Microbiol 98: 1073-1088.

Kastle, B., T. Geiger, F.L. Gratani, R. Reisinger, C. Goerke, M. Borisova, C. Mayer & C. Wolz, (2015) rRNA regulation during growth and under stringent conditions in Staphylococcus aureus. Environ Microbiol 17: 4394-4405.

Munzenmayer, L., T. Geiger, E. Daiber, B. Schulte, S.E. Autenrieth, M. Fraunholz & C. Wolz, (2016) Influence of Sae-regulated and Agr-regulated factors on the escape of Staphylococcus aureus from human macrophages. Cell Microbiol 18: 1172-1183.

---

## [Author Response]

*Essential revisions:*

*1) In Figure 3 the authors use cell lines which they suggest have altered expression of TagB and therefore altered the teichoic acid levels of the bacterial cell wall. It does not appear that these strains have been previously published, and therefore it is important that the authors show that these strains do in fact have altered expression of TagB (either through qRT PCR or protein level analysis) and/or altered teichoic acid levels in the cell wall. This is particularly important as these strains are used in the final animal experiments, where a great deal of significance of the paper lies.*

This revised version includes a new panel in Figure 3—figure supplement 1, showing qRT-PCR analysis of *tagB* relative expression in the WT strain, the strain harboring the empty vector and the high-*tagB* and low-*tagB* strains that show altered teichoic acid levels. The low-*tagB* strain shows significantly lower *tagB* relative expression levels whereas high-*tagB* strain shows significantly higher *tagB* expression level in comparison to WT strain. This is explained in subsection “Increase in cell wall rigidity activates σ^B^,repressing the *agr* positive feedback loop” of this revised manuscript.

*2) The pivotal finding of the paper is that in different sites of infection within the animal model S. aureus cells are in either the BRcell or DRcell state. As such, Figure 7 is critical to supporting this claim and would be improved with additional examples and clearer labeling of immune cell infiltrates. The low magnification images shown in Figure 7) do not contribute importantly to comprehension of the figure, and do not seem to highlight immune cell infiltrates. Including additional images in this figure (or as a supplement) displaying this is recommended. Additionally, it is unclear exactly how the authors performed the "semi-quantitative" imaging of the BRcell and DRcell reporter cells. It states that it was performed on three images, were these from independent animals? The same kidney or multiple infected kidneys? Additional explanation of this quantification method and representative fluorescence images are warranted here as well.*

We have followed these referee’s recommendations and included the following modifications in this revised manuscript:

1) Additional microscopy images of infected tissues are included in Figure 7—figure supplement 1. This figure shows detailed images of immune cell infiltrates, which have been more clearly identified using different labelling techniques (Giemsa and Eosin-Hemotoxylin) (Thammavongsa et al., 2009). In addition, 10x microscopy images of Figure 7 have been moved to Figure 7—figure supplement 3. The purpose of low-magnification images is to document the formation of *S. aureus* aggregates in infected kidney, which are not observed in infected heart. We have clarified this better in subsection “*BR* and *DR* cell fates arise during in vivo infections”.

2) We expanded the Materials and methods section and explained more clearly the procedure for the quantification of *S. aureus* fluorescence in infected tissues using confocal microscopy. We have included additional references in which a similar approach has been used. In infected tissues, the formation of bacterial aggregates make a clear definition of single cells difficult. Thus, we used a cell imaging approach to quantify fluorescence, by calculating the image area of the bacterial aggregate that shows fluorescence relative to the total area of the bacterial aggregate. Using this approach in a number of technical and biological replicates enabled us to obtain a percentage of the fluorescence bacterial subpopulation in the total population.

3) In the Materials and methods section, we clarified that animal infection analyses were performed using three different images, each one obtained from organs of different infected animals (n = 3 biological replicates). From each sample, we compared three different areas of the infected organ (technical replicates) to verify that selected samples were representative of the infection stage of the organ. In this revised manuscript, we have included examples of biological replicates from different infected organs (Figure 7—figure supplement 2), as the referees recommended.

*3) The overall organization of the results and figures made reading and interpreting the data more difficult than necessary. The results are not discussed in the order of the figure panels, and the supplemental figures were out of order in the manuscript. Organizing the figure panels in the order they are discussed in the Results section (or vice versa) will greatly improve the reading experience.*

We have reorganized the Results section to discuss data in the order of figure panels and supplemental figures to facilitate the reading.

*4) There are quite a number of publications dealing with heterogeneity and adaptation in S. aureus subpopulations and which regulators control heterogeneity – including the following. The findings should be placed into context.*

Audretsch, C., D. Lopez, M. Srivastava, C. Wolz & T. Dandekar, (2013) A semi-quantitative model of Quorum-Sensing in Staphylococcus aureus, approved by microarray meta-analyses and tested by mutation studies. Mol Biosyst 9: 2665-2680.

Geiger, T., B. Kastle, F.L. Gratani, C. Goerke & C. Wolz, (2014) Two small (p)ppGpp synthases in Staphylococcus aureus mediate tolerance against cell envelope stress conditions. J Bacteriol 196: 894-902.

Geiger, T. & C. Wolz, (2014) Intersection of the stringent response and the CodY regulon in low GC Gram-positive bacteria. Int J Med Microbiol 304: 150-155.

George, S.E., T. Nguyen, T. Geiger, C. Weidenmaier, J.C. Lee, J. Liese & C. Wolz, (2015) Phenotypic heterogeneity and temporal expression of the capsular polysaccharide in Staphylococcus aureus. Mol Microbiol 98: 1073-1088.

Kastle, B., T. Geiger, F.L. Gratani, R. Reisinger, C. Goerke, M. Borisova, C. Mayer & C. Wolz, (2015) rRNA regulation during growth and under stringent conditions in Staphylococcus aureus. Environ Microbiol 17: 4394-4405.

*Munzenmayer, L., T. Geiger, E. Daiber, B. Schulte, S.E. Autenrieth, M. Fraunholz & C. Wolz, (2016) Influence of Sae-regulated and Agr-regulated factors on the escape of Staphylococcus aureus from human macrophages. Cell Microbiol 18: 1172-1183.*

These references have been included in this revised manuscript and placed into context.